# Tracing LLM Reasoning Processes with Strategic Games: A Framework for Resource-Constrained Decision Making and Revision

## Abstract

Large language models (LLMs) are increasingly applied to tasks that require complex reasoning. While most benchmarks focus on evaluating final reasoning outcomes, they overlook the internal processes that lead to those outcomes—such as how a model revises and makes decisions under constraints. We argue that evaluating these internal reasoning steps is essential for understanding model behavior and improving reliability in real-world applications. To make these processes observable and measurable, we propose using strategic games as a natural and effective environment. These games operate within closed, rule-based systems and provide interpretable states, limited resources, and automatic feedback. Therefore, we propose a framework to evaluate LLMs along two core process dimensions: resource-constrained decision making and revision. To support this, we introduce a set of evaluation metrics that extend beyond traditional Win Rate (WR), incorporating measures such as Over-Correction Risk Rate (ORR), Correction Success Rate (CSR), Improvement Slope ($\beta$), and Over-Budget Rate (OBR). In a set of 4320 adversarial rounds across 12 state-of-the-art models, we find that ChatGPT-o3-mini, which demonstrates strong reasoning capabilities, achieves the strongest results across process metrics (74.7% WR, 78.6% CSR, and $\beta = +0.041$). In contrast, Qwen-Plus, despite a high ORR of 81.6%, achieves only a 25.6% win rate, primarily due to excessive resource use. We observe a consistent negative trend between ORR and CSR, suggesting that more frequent corrections do not always improve outcomes. This pattern may reflect high-frequency revisions made prematurely, which often reduce overall effectiveness, whereas more targeted revisions are associated with higher accuracy. We hope this work offers a new direction for LLM evaluation—focusing not just on what models decide, but on how they decide it.

## 1 Introduction

Large language models (LLMs) have demonstrated remarkable progress in solving increasingly complex reasoning tasks (Huang & Chang, 2023; Zhang et al., 2024b). These advances have enabled LLMs to perform impressively in real-world applications requiring reasoning and decision-making. To measure these abilities, a variety of benchmarks—such as GSM8K (Cobbe et al., 2021), MMLU (Hendrycks et al., 2020), and MBPP (Austin et al., 2021)—have been widely adopted. These benchmarks have played a central role in driving LLM development and tracking performance improvements over time (Chang et al., 2024).

However, as LLM performance on these benchmarks continues to improve, numerous studies have shown that relying solely on fixed problem sets and outcome-only evaluation presents significant limitations. Most benchmarks reduce evaluation to fixed question sets and final-answer correctness, offering little insight into whether the reasoning process itself is valid, faithful, or consistent (Xia et al., 2025; Albrecht et al., 2022). Consequently, models may appear successful while in fact producing correct answers for the wrong reasons, obscuring fundamental flaws in their reasoning and neglecting the instability of reasoning dynamics (Banerjee et al., 2024; Varela et al., 2025). Moreover, fixed datasets are vulnerable to data contamination, which makes it difficult to disentangle genuine reasoning ability from memorization effects (Zhang et al., 2024a). While some recent

studies attempt to address these issues, such as using automatically generated questions (Yu et al., 2024), these approaches introduce new challenges, including unstable difficulty and occasional invalidity (Minderer et al., 2021). Therefore, we argue that incorporating process-level evaluation is a necessary next step in LLM evaluation.

Building on this motivation, we propose shifting the evaluation paradigm: from static, outcome-based tests toward dynamic, process-aware environments. We identify **strategic games** as a particularly well-suited testbed for this purpose. Games provide closed, rule-based environments with clear feedback signals, bounded resources, and interpretable decision traces (Duan et al., 2024; Gui et al., 2024). Their structure allows us to directly observe and quantify multi-step reasoning behaviors—without requiring human annotations or handcrafted evaluation rubrics.

In this work, we introduce **AdvGameBench**, an evaluation framework that leverages adversarial game environments and incorporates dynamic assessment metrics. Rather than judging success solely by win rates, our framework traces how models adapt their strategies, revise them when needed, and operate under strict resource budgets. We define two core evaluation dimensions—**resource-constrained decision making** and **revision**—and propose concrete metrics that capture each of them. By embedding these metrics into interactive adversarial environments, AdvGameBench extends beyond fixed datasets and outcome-based evaluation by providing a scalable method to more directly observe reasoning behaviors without requiring additional human annotations. This enables researchers to systematically analyze how models adapt strategies, recover from errors, and make trade-offs under constraints, thereby offering a more fine-grained and reliable characterization of their reasoning capabilities.

To support broad and interpretable analysis, AdvGameBench spans three classic game genres—tower defense, battle card , and turn-based combat—each selected to represent distinct reasoning and strategic demands. The framework logs full model outputs and action traces, enabling detailed inspection of decision quality, revision behavior, and adherence to constraints.

Our key contributions are:

- Introduce dynamic, multi-round, and process-aware metrics for evaluating LLMs in adversarial settings, moving beyond simple win/loss outcomes.
- Decompose reasoning into two dimensions—resource-constrained decision making and revision—and define metrics to capture them.
- Design and re-implement adversarial game environments to reduce the risk of data contamination and ensure fairness.

## 2 RELATED WORK

**LLMs in Gaming Applications.** LLMs have rapidly evolved and demonstrated significant capabilities across various complex tasks, including gaming scenarios (Sudhakaran et al., 2023; Xu et al., 2023; Gupta, 2023; Nananukul & Wongkamjan, 2024; Light & et al., 2023; Mathieu et al., 2023) and other strategic interactions like bargaining, cooperation, and decision-making (Xia et al., 2024; Mosquera et al., 2024; de Curtò et al., 2023). Early work explored text-based adventure games, including interactive fiction (Tsai et al., 2023) and two-player settings like the Prisoner's Dilemma (Akata et al., 2023), which revealed cooperation and coordination strengths. More recently, research has shifted to multiplayer card games such as Guandan (Yim et al., 2024), where Theory-of-Mind prompting improved collaboration but exposed weaknesses in long-horizon state management.

**Existing Benchmarks for LLM Evaluation.** A variety of benchmarks have been developed to evaluate LLM reasoning, ranging from planning tasks to interactive and game-based settings. Some benchmarks focus on planning tasks like PlanBench (Valmeekam et al., 2023), TravelPlanner (Xie et al., 2024), and ACPBench (Kokel et al., 2025), others evaluate LLMs in game-like settings. For instance, AppWorld (Trivedi et al., 2024), GTBench (Duan et al., 2024), GAMEBENCH (Costarelli et al., 2024), GameTraversalBenchmark (Nasir et al., 2024), MINT (Wang et al., 2023), AgentBench (Liu et al., 2023), and MT-Bench (Zheng et al., 2023) illustrate established efforts focusing on puzzle, multi-turn interactions, or agent-oriented tasks. Furthermore, Yang et al. (2023) provided benchmarks specifically for StarCraft II, showcasing sophisticated summarization techniques in strategic gaming contexts. Another research direction evaluates strategic reasoning using game-theoretic

frameworks, demonstrating how sophisticated models like GPT-4 (OpenAI, 2024) approximate human decisions, these studies remain focused on outcome-level metrics rather than the underlying reasoning processes.

**Key Difference.** Prior benchmarks emphasize outcome accuracy on fixed task collections, limiting evaluation to static correctness. In contrast, our benchmark places models in strategic, rule-based adversarial environments where processes are directly observable, shifting the focus from outcome accuracy to how models adapt and revise under constraints.

# 3 METHOD

## 3.1 MULTI MODEL ADVERSARIAL STRUCTURE

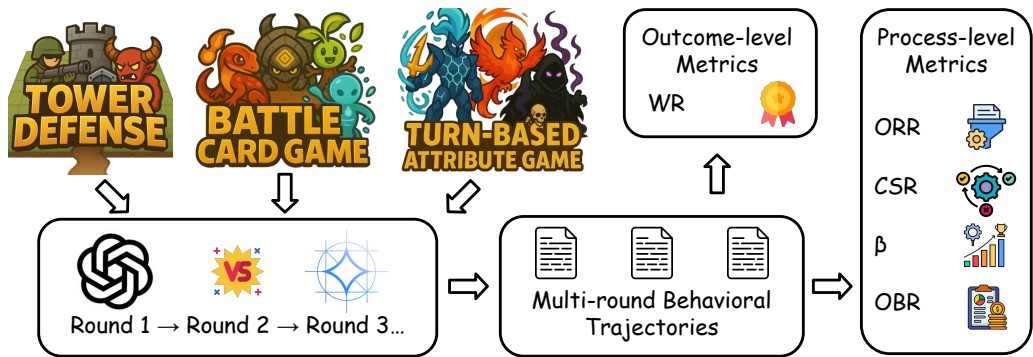

Figure 1: This figure illustrates the AdvGameBench evaluation pipeline. Three strategic game genres—tower defense, battle card, and turn-based combat—form the core environments for model evaluation. In each round, the model generates a strategy based on explicit rules; the simulator executes the strategy and returns an outcome and this process iterates over multiple rounds. These interactions yield behavioral trajectories from which process-level metrics are computed, including win rate (WR), over-correction risk rate (ORR), correction success rate (CSR), improvement slope ($\beta$), and over-budget rate (OBR).

We introduce **a structured adversarial framework** that goes beyond outcome-based evaluation and enables process-level assessment of LLM reasoning, specifically focusing on resource-constrained decision making and revision in rule-governed environments. Each evaluation is conducted in a game-based, closed-loop setting governed by explicit rules and resource constraints. The models receive identical prompts, independently generate strategies, and the simulator executes these strategies to produce a rule-verifiable win/loss outcome.

To ensure balanced evaluation, we construct three adversarial games—tower defense, battle card, and turn-based combat—each designed to target a distinct reasoning capability. In every round, models alternate between attacker and defender roles, and each model pair is evaluated under both move orders. Together, these conditions yield a dense adversarial match matrix covering all model pairs, roles, and move orders, enabling systematic comparison of constraint adherence and revision dynamics under matched conditions.

Beyond outcomes, our framework supports process-level analysis. After each round, models receive outcome-based feedback and may optionally revise their strategies. These revision sequences are logged and scored using process-level metrics including correction success rate, over-correction risk, and improvement slope. In addition, we measure constraint adherence through the over-budget rate, which quantifies whether models comply with explicit resource limits.

All game environments are completely re-implemented with shifted design from popular games to avoid strategy leakage from those games. This ensures that models cannot rely on memorized heuristics or latent familiarity with existing game patterns, preserving the objectivity of the evaluation.

## 3.2 GAME SUITES: TOWER DEFENSE, BATTLE CARD, TURN-BASED

To evaluate how LLMs revise and adapt across varied reasoning contexts, we design three game environments that span distinct forms of strategic complexity. Each environment imposes different

constraints and interaction patterns: Tower Defense emphasizes spatial planning under sequential threats; Battle Card requires resource allocation and composition under outcome uncertainty; and Turn-based tests decision consistency across multi-step attribute interactions. This diversity ensures that our evaluation covers a broad spectrum of process-level reasoning behaviors.

**Tower Defense Game.** In this environment, models alternate between attacker and defender roles. Defenders place units on an 11-column battlefield to block attackers advancing from the right. Attackers aim to reach the left boundary, while defenders strive to destroy all attackers. Success criteria and rule violations provide clear feedback for iterative strategy refinement (see A.1).

**Battle Card Game.** Models control units with distinct attributes: attackers prioritize damage, defenders emphasize protection and recovery. Units engage in automated battles, with combat sequence determined by the number of units each side possesses. The side that eliminates all opposing units first wins, offering explicit outcome-based feedback for model improvement (see A.2).

**Turn-based Attribute Game.** Each side controls three characters with assigned elemental attributes (Fire, Wood, Water, Earth, Light, Dark), featuring strategic interactions based on attribute strengths and weaknesses. Characters choose three skills within a budget constraint, cycling through them in combat. Duels continue until one side remains, clearly indicating the strategic effectiveness and compliance of each model's choices (see A.3).

### 3.3 Evaluation Metrics

**Win Rate (WR).** WR measures the proportion of matches a model wins out of all played games, with rule violations resulting in immediate forfeiture. This metric captures the final outcome of the reasoning process and provides a baseline for comparison. It reflects how effectively a model transforms reasoning into an executable solution under rule constraints.

**Over-Correction Risk Rate (ORR).** ORR captures how frequently a model reacts to negative feedback with a revised proposal. This metric targets a critical behavior: over-adjustment in response to failure signals. In practical settings, excessive self-editing can reduce decision stability and degrade coherence over long horizons. High ORR indicates a lack of strategic confidence or an overly reactive revision policy. The need to track this behavior is grounded in the observation that models can degrade their own solutions through unnecessary changes, even when initial strategies are viable.

**Correction Success Rate (CSR).** CSR measures how often a revision leads to an improved result—either by eliminating a rule violation or by turning a loss into a win. This metric isolates the effectiveness of the model's internal feedback loop: can it not only detect failure but also recover from it? A model that frequently edits without reliably improving demonstrates superficial adaptivity rather than meaningful self-correction.

**Improvement Slope ($\beta$).** $\beta$ captures whether a model improves over repeated interactions in matched environments. This measures whether the model can adapt its strategy based on prior failures against a fixed opponent type. Unlike static metrics, $\beta$ traces whether a model learns or degrades over time. A flat or negative slope suggests overfitting or myopic adjustment; a positive slope reflects effective cumulative reasoning.

**Over-Budget Rate (OBR).** OBR measures how often a model generates proposals that exceed explicit resource constraints. This metric directly evaluates a model's ability to integrate symbolic or numerical limits into its reasoning process. Many LLMs can optimize performance under unconstrained conditions, but OBR reveals whether they can internalize hard boundaries and behave accordingly. This behavior is essential for real-world deployment, where compliance with external rules is not optional but required for safe execution.

Further detailed metrics are discussed in Appendix B.

## 4 Results

We evaluate 12 leading LLMs, including DeepSeek-R1/V3 (DeepSeek-AI et al., 2024), Qwen-Plus/Max (Bai et al., 2023), Claude-3.5-Sonnet (Anthropic, 2024), ChatGPT-4.1/4o/o3/o3-mini (OpenAI, 2024), Gemini-2/2.5-Flash (Anil et al., 2025), and LLaMA-3-70B (Grattafiori et al., 2024). To assess robustness, each model was tested against three diverse opponents: ChatGPT-4o,

Claude-3.5-Sonnet, and DeepSeek-V3. This setup avoids evaluation bias caused by shared architectures or training data. In each round, models play against all opponents in turn-based games, with the platform logging win/loss results and correction behaviors for downstream analysis.

## 4.1 REVISION BEHAVIOR: CORRECTION RATE & SUCCESS

| Model | TDG | | | BCG | | | TAG | | | avg | | |
|---|---|---|---|---|---|---|---|---|---|---|---|---|
| | WR | ORR | CSR | WR | ORR | CSR | WR | ORR | CSR | WR | ORR | CSR |
| ChatGPT-4.1 | 45.0 | 85.7 | 40.0 | 52.5 | 69.7 | 65.2 | 57.5 | 82.4 | 67.9 | 51.7 | 79.4 | 56.8 |
| ChatGPT-4o | 65.8 | 81.8 | 55.6 | 60.8 | 44.0 | 63.6 | 59.1 | 82.4 | 46.4 | 58.6 | 70.4 | 52.6 |
| ChatGPT-o3 | 75.8 | 41.1 | 57.1 | 76.7 | 50.0 | 88.9 | 70.0 | 30.0 | 66.7 | 74.2 | 40.0 | 73.7 |
| ChatGPT-o3-mini | 63.3 | 25.9 | 57.1 | 74.2 | 31.6 | 100.0 | 86.7 | 9.0 | 100.0 | 74.7 | 24.5 | 78.6 |
| Claude-3-5-Sonnet | 56.7 | 89.3 | 56.0 | 45.8 | 70.0 | 64.3 | 55.0 | 76.9 | 65.0 | 52.5 | 77.7 | 61.6 |
| DeepSeek-R1 | 70.8 | 53.6 | 80.0 | 49.2 | 32.2 | 40.0 | 80.0 | 83.3 | 70.0 | 66.7 | 48.4 | 63.3 |
| DeepSeek-V3 | 43.3 | 84.6 | 45.5 | 23.3 | 75.5 | 24.3 | 56.7 | 75.0 | 38.1 | 41.1 | 78.4 | 35.2 |
| Gemini-2-Flash | 15.8 | 90.6 | 10.4 | 49.2 | 65.7 | 60.9 | 38.3 | 67.5 | 28.0 | 34.4 | 76.8 | 27.1 |
| Gemini-2.5-Flash | 60.0 | 40.0 | 60.0 | 59.0 | 79.2 | 68.4 | 58.1 | 76.2 | 56.3 | 62.5 | 65.7 | 63.0 |
| LLaMA-3-70B | 33.3 | 90.2 | 29.7 | 42.5 | 76.3 | 51.7 | 65.0 | 69.2 | 66.7 | 46.9 | 80.0 | 45.2 |
| Qwen-Max | 39.2 | 44.7 | 5.8 | 10.8 | 50.0 | 10.3 | 41.7 | 51.3 | 36.9 | 30.5 | 48.9 | 16.9 |
| Qwen-Plus | 19.2 | 78.4 | 20.0 | 16.7 | 81.5 | 13.6 | 40.8 | 86.1 | 45.2 | 25.6 | 81.6 | 24.3 |

Table 1: Performance of models across three adversarial game environments (TDG, BCG, TAG), measured by WR, ORR, and CSR.

**Table 1** reports WR, ORR, and CSR of each model across the three game environments, together with their averages. ChatGPT-o3-mini and ChatGPT-o3 attain the highest WR (74.7% and 74.2%), while the Qwen family and Gemini-2-Flash underperform. This contrast indicates stronger outcome-level performance under identical rule constraints; to understand why, we next examine the revision metrics (ORR/CSR) and the constraint-adherence metric (OBR).

Revision behavior reveals a frequency–effectiveness gap: Qwen-Plus, DeepSeek-V3, and Claude-3.5-Sonnet exhibit high ORR, indicating frequent adjustments with limited stabilization. Such blind revisions occur even when the model lacks a clear understanding of where the error lies, reflecting a tendency to revise reactively in response to failure signals rather than making targeted corrections. In contrast, ChatGPT-o3-mini maintains a relatively low ORR (49.3%) coupled with the highest CSR (78.6%), indicating that its feedback response is well-calibrated: it avoids hasty revisions when the source of error is unclear, thereby maintaining greater decision stability. This can be regarded as a form of procedural robustness. This pattern holds across environments—ChatGPT-o3-mini reaches 100% CSR in BCG and TAG—while several weaker models revise aggressively in all games yet rarely turn those edits into wins. Across models, ORR tends to correlate negatively with CSR and win rate, reinforcing that frequent revisions often reduce stability rather than improve accuracy. Notably, Qwen-Max and Qwen-Plus revise often but succeed only around 20% of the time, reflecting a reactive but ineffective correction policy.

These findings underscore a broader principle: revision frequency alone is not a reliable indicator of performance. Effective models demonstrate selective correction, revising less often but with greater impact, whereas weaker models over-correct without corresponding gains. Thus, revision should be viewed not merely as responsiveness to feedback, but as a calibrated process whose success depends on aligning corrective actions with contexts that maximize the probability of improvement.

## 4.2 RESOURCE-CONSTRAINED DECISION MAKING

**Figure 2** reports the Over-Budget Ratio (OBR), which quantifies the proportion of turns in which a model exceeds the environment's resource constraints. While most models stay within budget in over 80% of turns, the variation across models is notable. ChatGPT-o3 and ChatGPT-o3-mini maintain perfect budget adherence, never exceeding the allowed limits. In contrast, Qwen-Plus surpasses its budget in approximately half of its turns, and Qwen-Max records similarly high overuse (OBRs of 0.50 and 0.45, respectively). This pattern is strongly aligned with performance: the o3 series models not only exhibit the lowest OBRs but also achieve the highest win rates (74.7% and

74.2%), whereas the Qwen models, with the highest OBRs, perform worst in terms of win rate (30.5% and 25.6%).

We further find a strong negative correlation between OBR and win rate (Pearson r = –0.95, $p < 0.001$), indicating that effective resource management is closely tied to model success. High OBRs are often associated with reactive, post-hoc revisions—corrections made after poor initial decisions—which typically fail to compensate for early mistakes. Conversely, models with low OBRs demonstrate more disciplined and efficient execution, avoiding costly errors in the first place. These results position OBR as a meaningful process-level indicator that goes beyond outcome accuracy, revealing how well models translate abstract constraints into concrete and consistent decision-making. Strong performers not only remain within budget but also allocate their resources strategi-

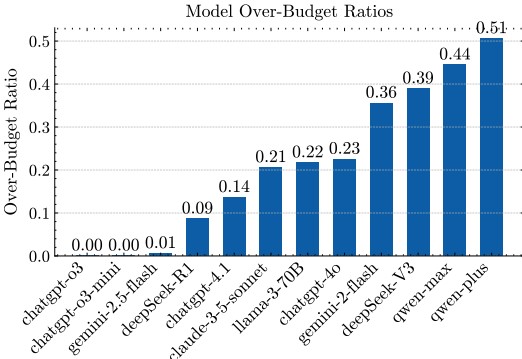

Figure 2: Over-Budget Ratio for Each Model

cally, contributing to higher correction success and overall coherence in behavior. In this sense, OBR serves as both a measure of resource discipline and a proxy for constraint adherence, signaling whether a model can deliver reliable performance within structural boundaries.

## 4.3 Init-win & Improvement Slope

We evaluate adaptation capabilities using two complementary metrics: initial win rate (init-win), which reflects first-round performance without feedback, and improvement slope, which measures a model's ability to enhance its strategy over time. Together, they capture a model's capacity to start strong and adapt through interaction. **Table 2** shows initial win rates and improvement slopes across models. DeepSeek-R1 achieves the highest init-win (75.0%) but declines over time, suggesting limited stability. In contrast, ChatGPT-o3 and o3-mini start with lower win rates (58.3%) yet steadily improve, reflecting stronger adaptability. Models like Gemini-2.5-Flash and Claude-3.5-Sonnet perform well initially but regress, likely

| Model | Init-win (%) | Slope |
|---|---|---|
| ChatGPT-4.1 | 47.2 | 0.0238 |
| ChatGPT-4o | 63.9 | 0.0087 |
| ChatGPT-o3 | 58.3 | 0.0452 |
| ChatGPT-o3-mini | 58.3 | 0.0413 |
| Claude-3.5-Sonnet | 55.6 | -0.0198 |
| DeepSeek-R1 | 75.0 | -0.0134 |
| DeepSeek-V3 | 52.8 | -0.0444 |
| Gemini-2-Flash | 38.9 | -0.0008 |
| Gemini-2.5-Flash | 69.4 | -0.0191 |
| LLaMA-3-70B | 55.6 | -0.0095 |
| Qwen-Max | 47.2 | -0.0389 |
| Qwen-Plus | 30.5 | 0.0024 |

Table 2: Initial win rates and improvement slopes across models.

due to static heuristics. Qwen models show little progress, pointing to weak utilization of feedback. Across families, only ChatGPT models consistently improve, reflecting stronger adaptation. These patterns demonstrate that robust performance depends on both strong initial outcomes and adaptability across repeated evaluations—a critical aspect quantified by process-level metrics such as the improvement slope.

To better understand these trends, we examine what the improvement slope captures beyond raw performance changes. The improvement slope reflects more than just numerical change; it also reveals whether a model can effectively leverage feedback and sustain long-term strategy optimization. This aspect is particularly critical in real-world scenarios, where models must continually interact and adjust. A high initial win rate alone is not sufficient to ensure robust performance—what ultimately determines long-term reliability is the ability to improve iteratively.

## 4.4 Role symmetry and first-move bias

We use First-Mover Advantage (FMA) to examine how model performance differs when initiating an action versus responding to a prior move. We analyze this effect across three dimensions: win rate, over-correction risk rate, and correction success rate. As shown in **Table 3**, most models exhibit relatively minor differences in win rate between first- and second-mover roles, with FMA values generally within five percentage points. This suggests limited systematic advantage based on turn order for overall success. However, several models deviate from this trend. Gemini-2-Flash (FMA = +13.2%) performs substantially better when acting first, while ChatGPT-4o (FMA = –21.7%)

| Model | FMA$_{win}$ | FMA$_{over-correction}$ | FMA$_{succ}$ |
|---|---|---|---|
| ChatGPT-4.1 | 0.067 | 0.137 | 0.200 |
| ChatGPT-4o | -0.217 | 0.148 | -0.004 |
| ChatGPT-o3 | 0.061 | 0.117 | -0.133 |
| ChatGPT-o3-mini | 0.017 | 0.096 | -0.083 |
| Claude-3-5-Sonnet | 0.117 | 0.056 | -0.064 |
| DeepSeek-R1 | -0.089 | -0.160 | -0.116 |
| DeepSeek-V3 | -0.111 | -0.094 | 0.099 |
| Gemini-2-Flash | 0.133 | 0.111 | 0.125 |
| Gemini-2.5-Flash | 0.061 | -0.097 | 0.212 |
| LLaMA-3-70B | 0.050 | 0.008 | -0.026 |
| Qwen-Max | -0.167 | -0.004 | 0.109 |
| Qwen-Plus | 0.022 | 0.139 | 0.032 |

Table 3: First-mover advantage (FMA) for WR, ORR, and CSR. FMA = first − second.

and Qwen-Max (FMA = –16.7%) exhibit the opposite pattern, achieving higher win rates when playing second. These results suggest that certain models are more sensitive to the structural asymmetries introduced by move order.

Stronger patterns emerge when examining correction behavior. As shown in Table 3, we observe that most models show a clear preference for initiating rather than responding. For example, ChatGPT-4o and ChatGPT-4.1 demonstrate significantly higher over-correction risk rates when acting first (FMA = +14.8% and +13.8%, respectively). Similarly, first-mover performance gains are evident in correction success rates for Gemini-2.5-Flash (+21.2%), Gemini-2-Flash (+12.5%), and ChatGPT-4.1 (+19.9%). These findings underscore the importance of accounting for role asymmetry in evaluation setups. Our dual-first configuration helps mitigate first-mover bias, offering a more balanced and interpretable view of model behavior under asymmetric game dynamics.

## 4.5 DOES REVISING MORE REALLY HELP?

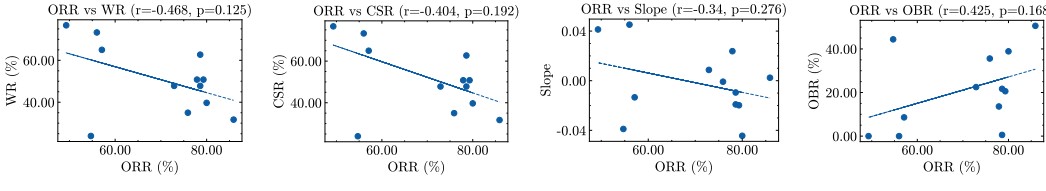

(a) Correlation between ORR and WR

(b) Correlation between ORR and CSR

(c) Correlation between ORR and Slope

(d) Correlation between ORR and OBR

Figure 3: Correlation between ORR and four process-level metrics (WR, slope, OBR, CSR) across models. Higher ORR values align with poorer outcomes: lower WR, slower slope, higher OBR, and lower CSR.

We quantify a model's tendency to revise reactively using the ORR—the probability that a model submits a new strategy immediately after receiving explicit negative feedback. **Figure 3** presents the correlation between ORR and four process-level outcomes. We observe a moderate negative relationship between ORR and final win rate ($r = -0.47$, $p = 0.13$), suggesting that models which revise more frequently tend to achieve lower overall success. Similarly, ORR correlates negatively with improvement slope ($r = -0.34$, $p = 0.28$), indicating that frequent edits do not accelerate strategic refinement. In terms of budget use, models with higher ORR values are more likely to exceed resource constraints (OBR; $r = +0.43$, $p = 0.17$), and also show lower correction success rates ($r = -0.34$, $p = 0.28$), implying that high-frequency revision may undermine the quality of attempted corrections.

Given the sample size ($n = 12$), these correlations do not reach statistical significance. However, the directional consistency across all metrics is notable. Models with high over-correction risk consistently perform worse on efficiency and resource discipline. In contrast, top-performing models

like ChatGPT-o3-mini combine low ORR with high correction success. These observations suggest that effective strategy adjustment tends to rely on precision rather than high-frequency revision.

### 4.6 HOLISTIC COMPARISON VIA RADAR CHART

To synthesize model performance across reasoning dimensions, we constructed a radar chart visualizing five normalized metrics: win rate (WR), correction success rate (CSR), improvement slope, 1 − over-correction risk rate (ORR), and 1 − over-budget rate (OBR). All metrics were scaled to a common range, with inversions applied where necessary so that higher values consistently indicate better performance. This unified view enables a comparative assessment of both outcome and process quality across models.

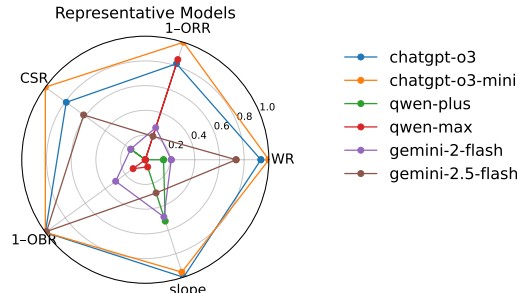

Figure 4: Radar chart of model performance across five normalized metrics (WR, CSR, slope, 1 − ORR, 1 − OBR ). Larger and more balanced areas, as in ChatGPT-o3 and o3-mini, reflect robust reasoning pipelines, while imbalances in Qwen and Gemini highlight that balanced outcomes and constraint adherence matter more than excelling in any single metric.

**Figure 4** presents the radar chart, where ChatGPT-o3 and o3-mini form the largest areas, reflecting strong and consistent performance across all dimensions. This combination of high win rates, effective corrections, stable improvement, and disciplined resource use suggests a well-integrated reasoning pipeline. In contrast, models like Qwen-Plus and Qwen-Max show sharp imbalances, marked by frequent but ineffective revisions and frequent budget violations. Gemini models reach moderate correction success but remain limited by elevated correction risk or weaker budget control. These representative cases illustrate that top performance depends on balanced outcomes, effective revisions, and adherence to constraints, rather than strength on a single metric.

Importantly, our aim is not to diminish the role of win rate, which remains a critical outcome-level measure. Rather, the process-level metrics complement win rate by revealing why a model succeeds or fails, offering explanatory context for differences in final outcomes.

Larger radar areas correspond to more robust reasoning pipelines, underscoring that process quality is essential to understanding model competence. The complete set of radar plots for all evaluated models is provided in Appendix A.6.

## 5 DISCUSSION

**Empirical studies.** Our study of twelve production-scale LLMs across **4 320** adversarial rounds yields three consistent findings:

1. **Integrated skill trumps single metrics.** Models that balance across multiple dimensions—notably CHATGPT–O3-MINI with a 74.7 % win rate, 78.6 % correction-success rate, and positive improvement slope of +0.0413—outperform models that excel in only one aspect.
2. **"Spray-and-pray" revision is counter-productive.** QWEN-PLUS issues corrections in 81.6 % of error states yet wins only 25.6 % of games and overspends in nearly half the turns. Across all systems, correction frequency and efficacy are negatively correlated (Pearson $r = -0.51$, $p = 0.093$), indicating that *calibrated* self-editing matters more than sheer persistence.

**Hallucination.** In our tower defense experiments, all defensive units were consistently referred to as *soldiers*. However, several models frequently generated the term *peashooter*, which was never introduced in the task instructions. A review of the interaction logs reveals that this phenomenon is more likely to originate from pretraining associations—specifically, the frequent co-occurrence of "tower defense" and the game Plants vs. Zombies in web-scale corpora. This leads models to reproduce familiar terminology, even when it conflicts with the defined rules of the environment. Such behavior undermines the validity of the benchmark, effectively turning the evaluation into a

test of memorized correlations rather than genuine adaptation to constraints. To eliminate this form of memory bias, a challenge addressed by contamination-free benchmarks (White et al., 2024), we redesigned the game environment to neutralize lexical cues and ensure that performance reflects models' ability to engage with novel rules and dynamic constraints, rather than recalling pretraining artifacts.

To further examine this effect, we conducted two complementary tests with ChatGPT-4o. In the first, when provided only with the positions of zombies and no additional information, the model spontaneously generated plants from Plants vs. Zombies, with well-aligned coordinates despite the absence of task instructions. In contrast, under the same zero-information setting in our redesigned tower defense game, ChatGPT-4o failed to produce any coherent or rule-compatible output. To further validate this observation, we conducted 50 fixed-level trials using prompt substitution. When given only example input without any rule or role information, the model achieved a 30% pass rate in Plants vs. Zombies, yet failed completely in our redesigned tower defense game. Together, these results suggest that the use of peashooter and related terms is more likely to originate from pretrained associations rather than genuine reasoning, and underscore the effectiveness of our contamination-resistant design for fair evaluation.

**Limitations.** AdvGameBench currently (i) focuses on three turn-based genres rather than attempting to span all possible real-time or cooperative formats, and (ii) evaluates a representative set of 12 mainstream LLMs instead of exhaustively including smaller or multimodal variants. These scope choices simplify evaluation and ensure reproducibility, while still being broad enough to demonstrate the utility and generality of our process-level metrics.

# 6 CONCLUSION

Static accuracy benchmarks often cannot explain why a model produces correct or incorrect answers, nor can they reveal how these results are generated. AdvGameBench provides an open and extensible environment in which such process-level traits can be systematically observed and quantified through complete decision traces.

Win rate remains the most direct and critical outcome-level measure, but by itself it is insufficient to uncover the essence of model competence. For example, models with similar win rates may differ substantially in their reasoning strategies, revision habits, and resource management. In such cases, process-level metrics serve as a complementary lens, revealing how a model gradually moves toward success or failure, whether through disciplined revision strategies or effective budget management.

Our work aims not only to ask "Did the model win?" but also to explain "Why did it win?" and "Was the victory stable or coincidental?" Win rate and process-level metrics are not competing measures but complementary signals that together yield a more complete and interpretable evaluation framework. We hope that AdvGameBench can help drive a shift in LLM evaluation: from merely asking "Was the answer correct?" to further asking "Under what constraints, and through what processes, was the answer produced?"

# 7 ETHICS STATEMENT

This work does not involve human subjects, personally identifiable information, or sensitive data. All experiments were conducted on publicly available large language models within synthetic game environments. The study raises no direct ethical concerns regarding privacy, fairness, or safety.

# 8 REPRODUCIBILITY STATEMENT

We provide detailed descriptions of the game environments, evaluation metrics, and experimental protocols in the main paper. Additional implementation details, rules, unit attributes, and sample prompts are included in the appendix to facilitate replication of our experiments.

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

## USE OF LLMs

Large language models (LLMs), such as ChatGPT/GPT-4, were used solely for writing assistance, including polishing grammar, improving readability, and suggesting alternative phrasings. All re-search ideas, experimental design, and analyses were developed entirely by the authors.

## A APPENDIX

### A.1 EXTENDED EXPERIMENTAL RESULTS

### A.1.1 EXTENDED EVALUATION ON LATEST MODELS

Table 4: Extended leaderboard of 13 modern LLMs evaluated over 4,680 matches against strong opponents (ChatGPT-4o, Claude-4-Sonnet, DeepSeek-V3).

| Model Name | Win Rate (WR) ↑ | ORR ↓ | CSR ↑ | Slope ($\beta$) | OBR ↓ |
|---|---|---|---|---|---|
| GLM-4.5-Air | 44.2% | 0.217 | 0.190 | 0.011 | 0.471 |
| GLM-4.6 | 32.2% | 0.569 | 0.217 | -0.022 | 0.646 |
| Kimi-K2-Instruct | 43.9% | 0.751 | 0.423 | 0.019 | 0.738 |
| Qwen2.5-32B-Instruct | 29.1% | 0.685 | 0.243 | -0.022 | 0.896 |
| Qwen2.5-72B-Instruct | 46.8% | 0.344 | 0.375 | -0.030 | 0.804 |
| Qwen3-30B-A3B | 15.8% | 0.294 | 0.120 | 0.001 | 1.000 |
| Qwen3-Max | 44.2% | 0.724 | 0.360 | 0.010 | 0.620 |
| DeepSeek-V3.1 | 31.1% | 0.454 | 0.435 | -0.025 | 0.645 |
| Gemini-2.5-Flash-Lite | 52.9% | 0.746 | 0.045 | -0.014 | 0.568 |
| Gemini-2.5-Pro | 73.8% | 0.388 | 0.500 | 0.005 | 0.038 |
| Gemini-3-Pro | 79.0% | 0.086 | 0.000 | 0.009 | 0.000 |
| GPT-5 | 77.9% | 0.567 | 0.143 | -0.017 | 0.000 |
| GPT-5-Mini | 71.3% | 0.600 | 0.133 | -0.017 | 0.008 |

### A.1.2 STABILITY ANALYSIS ACROSS RANDOM SEEDS

Table 5: Stability analysis across random seeds ($N = 3$). Results are reported as Mean $\pm$ Standard Deviation.

| Model Name | Win Rate (WR) | ORR | CSR |
|---|---|---|---|
| GLM-4.5-Air | $0.106 \pm 0.011$ | $0.585 \pm 0.191$ | $0.127 \pm 0.070$ |
| Qwen2.5-72B-Instruct | $0.367 \pm 0.123$ | $0.324 \pm 0.143$ | $0.261 \pm 0.067$ |
| ChatGPT-4o | $0.561 \pm 0.011$ | $0.621 \pm 0.194$ | $0.409 \pm 0.031$ |
| ChatGPT-o3-mini | $0.614 \pm 0.021$ | $0.403 \pm 0.117$ | $0.467 \pm 0.176$ |
| Claude-4-Sonnet | $0.439 \pm 0.011$ | $0.568 \pm 0.327$ | $0.336 \pm 0.141$ |
| Gemini-2.5-Pro | $0.777 \pm 0.037$ | $0.414 \pm 0.240$ | $0.596 \pm 0.258$ |

### A.2 TOWER DEFENSE GAME

#### A.2.1 GAME RULES

1. Players can purchase characters and place them on the battlefield. The battlefield consists of 5 rows (corresponding to y-coordinates 0-4). The human side can place units in a designated area spanning 11 columns (corresponding to x-coordinates 0-10).
2. Demons spawn from the right side of the battlefield (x-coordinates 11) and move left. Human units are placed on the left side of the battlefield, remain stationary, and attack approaching enemies.
3. All units attack according to their attack interval, automatically attacking when their cooldown ends. Defending units fire bullets or activate skills to attack enemies. Invading units engage in melee attacks when they come into contact with defending units.
4. Each grid cell can only contain one human unit at a time. Placing a new unit in an occupied cell is not allowed.
5. When an attack hits, the target takes damage based on the attacker's power. If a unit's health drops to 0, it is eliminated and removed from the battlefield.
6. If all enemies are eliminated, the player wins. If any enemy successfully reaches the left side of the battlefield, the player loses.

#### A.2.2 HUMAN UNITS

| Unit | Attributes |
|---|---|
| HandgunSoldier | Health: 3, Shooting interval: 1000ms, Cost: 100, Damage per shot: 1, No special abilities. |
| RifleSoldier | Health: 3, Shooting interval: 500ms, Cost: 200, Damage per shot: 1, No special abilities. |
| MachineGunSoldier | Health: 3, Shooting interval: 250ms, Cost: 400, Damage per shot: 1, No special abilities. |
| ShieldSoldier | Health: 15, Cost: 50, Only for defense, no attack capabilities. |
| EnhancedShieldSoldier | Health: 30, Cost: 100, Only for defense, no attack capabilities and Bouncing Demon cannot jump over. |
| FlamethrowerSoldier | Health: 2, Cost: 200, Shooting interval: 1000ms, Damage per shot: 1, Deals an additional 1 damage every 1000ms. |
| IceSoldier | Health: 2, Shooting interval: 1000ms, Cost: 200, Damage per shot: 1, Reduces enemy speed by half. |
| AntiAirSoldier | Health: 2, Shooting interval: 1000ms, Cost: 175, Damage per shot: 1, Can attack airborne units. |

| Bomb | Health: 50, Detonation time: 500ms, Cost: 200, Explosion range: 3×3, Damage per explosion: 30, Destroyed after detonation. |
| LinearExplosion | Health: 50, Detonation time: 500ms, Cost: 200, Explosion range: the entire row, Damage per explosion: 30, Destroyed after detonation. |
| MagneticSoldier | Health: 2, Shooting interval: 2000ms, Cost: 100, Damage per shot: 0, Releases a magnetic pulse that disables the defensive abilities of ShieldDemon and MachineDemon. |
| LightMage | Health: 2, Damage per shot: 0, Cost: 150, No attack capabilities, Changes the attributes of bullets in the same row, converting their damage type to light. |
| RocketLauncherSoldier | Health: 2, Shooting interval: 1000ms, Damage per shot: 2, Cost: 600, Launches rockets, dealing damage to enemies within one grid. |

### A.2.3 DEMON UNITS

Note: A speed of 2 requires 14 seconds to travel from spawn to the last human grid.

| Unit | Attributes |
| --- | --- |
| NormalDemon | Health: 10, Speed: 2, Attack interval: 1000ms, Cost: 100, Damage per attack: 1, No special abilities. |
| GreatDemon | Health: 20, Speed: 2, Attack interval: 1000ms, Cost: 175, Damage per attack: 1, Higher health. |
| DemonKing | Health: 100, Speed: 2, Attack interval: 1000ms, Cost: 800, Damage per attack: 5. |
| SpeedyDemon | Health: 10, Speed: 4, Attack interval: 1000ms, Cost: 150, Damage per attack: 1, Moves faster. |
| ShieldDemon | Health: 10, Speed: 2, Attack interval: 1000ms, Cost: 175, Damage per attack: 1, Takes 70% less damage from normal attacks. |
| MachineDemon | Health: 20, Speed: 2 (increases to 3 when activated), Attack interval: 1000ms, Cost: 250, Damage per attack: 3, Reduced damage due to mechanical body. |
| BouncingDemon | Health: 10, Speed: 2, Attack interval: 1000ms, Cost: 150, Damage per attack: 1, Can jump over certain units except for the EnhancedShieldSoldier. |
| ShieldBreakerDemon | Health: 10, Speed: 2, Attack interval: 1000ms, Cost: 150, Damage per attack: 1 (×5 against shield units). |
| FireDemon | Health: 10, Speed: 2, Attack interval: 1000ms, Cost: 150, Damage per attack: 1, Immune to fire damage. |
| FrostDemon | Health: 10, Speed: 2, Attack interval: 1000ms, Cost: 150, Damage per attack: 1, Immune to ice damage and unaffected by slow effects. |
| FlyingDemon | Health: 10, Speed: 2, Attack interval: 1000ms, Cost: 200, Damage per attack: 1, Only affected by anti-air attacks and can pass through human units directly. |
| ShadowDemon | Health: 10, Speed: 2, Attack interval: 1000ms, Cost: 300, Damage per attack: 1, Can cast dark magic, making same-row allies immune to non-light damage. |
| SummoningDemon | Health: 10, Speed: 1, Attack interval: 1000ms, Cost: 300, Damage per attack: 1, Summons a Normal Demon to the left grid every 5000ms. |

## A.3 BATTLE CARD GAME

### A.3.1 GAME RULES

1. At the start of the game, players can purchase all desired characters at once, up to a maximum of 7 characters. Gold characters cost three times as much as bronze characters, but their stats (attack, health, numerical skill effects, etc.) are twice as high. Non-numerical skills are not affected by this multiplier.
2. Initiative Determination: The side with more characters attacks first. If both sides have the same number of characters, the invader attacks first.
3. Elemental Advantage: Certain elements have an advantage over others, granting a bonus in combat (Fire > Nature, Nature > Water, Water > Earth, Earth > Fire).
4. Battle Process: Both sides will attack based on their respective target_priority (target priority). However, if there are Taunt minions on the opponent's side, attackers must prioritize attacking them. The attack order follows a left-to-right sequence. The first minion in the invaders or defenders list (as defined in the JSON file) will attack first, depending on which side has the initiative. After that, the first minion from the opposing side attacks. Then, the second minion from the attacking side follows, then the second minion from the opposing side, and so on in an alternating pattern. If a minion's health reaches zero, it is eliminated. The battle continues with both sides attacking in turns until one side is completely wiped out, resulting in victory for the other side.
5. If all characters on one side are eliminated, the other side wins.
6. If both sides are eliminated simultaneously in the same attack resolution, the Invader wins.

### A.3.2 INVADER UNITS

| Unit | Attributes |
| --- | --- |
| FireLizard | Attack: 2, Health: 2, Cost: 1, Ability: Deals 2 damage to the enemy that killed it upon death. |
| WaterElemental | Attack: 2, Health: 2, Cost: 1, Ability: Gains +1 Attack when attacking. |
| PoisonFrog | Attack: 1, Health: 1, Cost: 2, Ability: Instantly destroys any minion it damages. |
| MoltenHound | Attack: 3, Health: 1, Cost: 2, Ability: Deals 1 damage to all enemies upon death. |
| BattleFrenzy | Attack: 7, Health: 4, Cost: 2, Ability: Each attack reduces its Attack by 4. |
| BanditLeader | Attack: 8, Health: 3, Cost: 3, Ability: Any excess damage from an attack carries over to the next target. |
| LavaGolem | Attack: 1, Health: 8, Cost: 3, Ability: Forces enemies to attack this minion first, Burns the attacker for 3 damage per turn when hit. |
| TideGuardian | Attack: 4, Health: 2, Cost: 3, Ability: Absorbs the first source of damage taken (divine shield), Attacks twice each turn. |
| TideLord | Attack: 4, Health: 9, Cost: 5, Ability: Doubles its Attack when taking damage. |
| Phoenix | Attack: 5, Health: 5, Cost: 5, Ability: Deals damage equal to its Attack to the target and its adjacent enemies, Revives with full Health after being defeated once per game. |
| ShadowOverlord | Attack: 4, Health: 4, Cost: 5, Ability: Summons a Slow Skeleton (3/1) upon death. |

### A.3.3 DEFENDER UNITS

| Unit | Attributes |
| --- | --- |

| | |
|---|---|
| Sapling | Attack: 2, Health: 2, Cost: 1, Ability: Gains +1 Health when attacking. |
| RockBeetle | Attack: 1, Health: 5, Cost: 1, Ability: Forces enemies to attack this minion before others. |
| ForestSeer | Attack: 2, Health: 2, Cost: 2, Ability: At the start of the game, grants +1 Attack and +2 Health to all Nature Allies. |
| StoneWarrior | Attack: 2, Health: 5, Cost: 2, Ability: Forces enemies to attack this minion before others. Summons a RockBeetle upon death. |
| EliteSoldier | Attack: 1, Health: 1, Cost: 2, Ability: At the start of the game, grants Divine Shield to adjacent minions and +1 Attack. |
| Paladin | Attack: 3, Health: 6, Cost: 3, Ability: Has Divine Shield; gains +2 Attack whenever a friendly minion loses its Divine Shield. |
| BlackRock | Attack: 5, Health: 1, Cost: 3, Ability: At the start of the game, gains +3 Health for each friendly minion. |
| VineProtector | Attack: 5, Health: 4, Cost: 3, Ability: Upon death, restores 2 Health to all friendly minions. |
| King | Attack: 3, Health: 10, Cost: 5, Ability: Summons a 2/2 Soldier with Divine Shield whenever it attacks (if there is an open space). |
| MountainGiant | Attack: 4, Health: 9, Cost: 5, Ability: Forces enemies to attack this minion first, Reduces the attack of the attacker by 2 when hit. |
| AncientTreant | Attack: 4, Health: 4, Cost: 5, Ability: At the start of the game, grants +3 Attack and +3 Health to all allied minions. |

## A.4 TURN-BASED ATTRIBUTE GAME

### A.4.1 GAME RULES

1. This game is a turn-based character battle game divided into two factions: Invader and Defender. Each faction consists of three characters. The Invader faction includes Fire, Water, and Dark elements, while the Defender faction includes Wood, Earth, and Light elements. Characters appear and act in the order they are listed in the data.
2. Combat proceeds in rounds. In each round, the three Invader characters act first in order, followed by the three Defender characters. The sequence then repeats in the next round.
3. Each character has three skills that are used in a preset, looping sequence. On each turn, a character uses the next skill in their list and continues cycling through them in order.
4. The game features an elemental effectiveness system: Fire beats Wood, Wood beats Earth, Earth beats Water, and Water beats Fire ($1.2\times$ damage when effective, $0.8\times$ when resisted). Light and Dark counter each other with $1.5\times$ damage. All other combinations deal the standard $1.0\times$ damage.
5. If all characters on one side are eliminated, the other side wins.

### A.4.2 INVADER SKILLS

| Skill Name | Description |
|---|---|
| **Fire Skills** | |
| flame_splash | Deals 12 damage and applies Burning for 2 rounds (1 layer, 5 damage per round). Cost: 1 |
| residual_warmth | Increases the damage of the next fire-based skill by 30% for 1 round. Cost: 1 |
| burst_flame_bomb | Deals 25 base damage, plus 3 additional damage for each Burning layer on the target. Cost: 2 |
| flame_whirlwind | Applies 4 layers of Burning to the target, lasting 2 rounds. Each layer deals 5 damage per round. Cost: 2 |

| | |
|---|---|
| magma_eruption | Deals 40 base damage, plus 5 extra damage per Burning layer. Removes all Burning after the attack. Cost: 3 |
| hell_curtain | Deals 35 damage and grants a shield that reflects 30 melee damage, lasting 2 rounds (1 layer). Cost: 3 |
| **Water Skills** | |
| stream_pierce | Deals 10 damage and grants 1 permanent layer of Tidal Surge. Cost: 1 |
| water_barrier | Grants a 5-point shield for 3 rounds and increases Tidal Surge by 1 layer. Cost: 1 |
| whirlpool_strangle | Deals 20 base damage, plus 4 additional damage per Tidal Surge layer. Cost: 2 |
| ice_branded | Deals 15 damage and causes the target to take 50% more damage next turn (1 round). Cost: 2 |
| tsunami_ending | Deals 30 base damage, plus 5 additional damage per Tidal Surge layer. Removes all Tidal Surge after the attack. Cost: 3 |
| abyss_resonance | Deals 3 damage per Tidal Surge layer and grants a shield worth 6 per layer, lasting 3 rounds. Cost: 3 |
| **Dark Skills** | |
| shadow_claw | Deals 14 damage and heals the user for 30% of the damage dealt (rounded down). Cost: 1 |
| fear_whisper | Reduces the target's damage taken by 10% for 3 rounds (1 layer). Cost: 1 |
| soul_siphon | Deals 25 damage. If the target's HP is below 50%, deals an extra 15 damage. Cost: 2 |
| night_ambush | Deals 20 damage and causes the target to take 20% more damage next turn (1 round). Cost: 2 |
| final_announcment | Deals 45 base damage, plus 5 extra damage for every 10% HP the target has lost. Cost: 3 |
| void_assimilation | Sacrifices 20% of current HP to deal penetration damage equal to twice the HP sacrificed. Cost: 3 |

### A.4.3 DEFENDER SKILLS

| Skill Name | Description |
|---|---|
| **Wood Skills** | |
| bud_healing | Grants Bud Healing for 3 rounds, restoring 6 HP per round. Cost: 1 |
| parasitic_seed | Applies Parasitic Seed for 3 rounds, immediately deals 10 damage. The target takes 5 counter damage each time they attack. Cost: 1 |
| life_totem | Restores 25 HP and grants Life Totem for 3 rounds, increasing healing received by 10%. Cost: 2 |
| natural_purification | Removes negative statuses from the user and deals 30 damage to the target. Cost: 2 |
| forest_reincarnation | Restores 60 HP. If it exceeds max HP, the excess is converted into a shield (50% of excess HP) for 3 rounds. Also deals 20 damage to an enemy. Cost: 3 |
| poison_vine | Applies Poison Vine for 3 rounds, dealing 25 damage per round. Cost: 3 |
| **Earth Skills** | |
| rock_armor | Grants 12 shield for 3 rounds and reflects 5 melee damage while the shield is active. Cost: 1 |
| earth_shock | Deals 20 damage. Cost: 1 |
| granite_barrier | Grants Granite Barrier for 3 rounds, decreasing damage by 40%. Cost: 2 |
| quicksand_trap | Applies Quicksand Trap for 3 rounds. The target's next 3 damage are delayed by 20% and each trigger deals 10 damage. Cost: 2 |
| earth_pulse | Grants shield based on HP lost (8 shield per 10% HP lost), lasting permanently. Cost: 3 |

| | |
|---|---|
| core_rebound | Deals 80% of stored damage to the target. Clears stored damage after use. Cost: 3 |
| **Light Skills** | |
| holy_glimmer | Removes a negative status (if any) and restores 8 HP to the user. Also deals 8 light damage to an enemy. Cost: 1 |
| faith_emblem | Grants Faith Emblem for 1 round. The next damage taken is reduced by 20% and converted into healing. When triggered, deals 10 damage to the attacker. Cost: 1 |
| divine_link | Grants Divine Link for 1 round. The next damage taken is reflected back to the attacker. Cost: 2 |
| luminous_dispel | Removes one buff from the target (if any) and applies a debuff for 2 rounds that reduces their attack by 15%. Cost: 2 |
| angelic_sanctuary | Grants Angelic Sanctuary for 3 rounds, reducing all incoming damage by 30 points. Cost: 3 |
| divine_sword | Deals 20 damage and grants a buff that increases the next skill's damage by 20. Cost: 3 |

## A.5 GAME PLACEMENT EXAMPLE

TDG:

```
{"placements": [{"unit": "HandgunSoldier", "x": 2, "y": 0},
                {"unit": "ShieldSoldier", "x": 1, "y": 0}],
 "total_cost": 150}
```

BCG:

```
{"roster": [{"name": "RockBeetle", "tier": "bronze"},
            {"name": "Paladin", "tier": "bronze"}],
 "order": [1, 2],
 "total_cost": 4}
```

TAG:

```
{"skills": {"fire": ["flame_splash", "residual_warmth",
↪  "burst_flame_bomb"],
           "water": ["stream_pierce", "water_barrier",
           ↪  "whirlpool_strangle"],
           "dark": ["shadow_claw", "night_ambush", "soul_siphon"]},
 "total_cost": 9}
```

## A.6 HOLISTIC COMPARISON VIA RADAR CHART (ADDITIONAL MODELS)

To synthesize performance for this *subset of additional models*, we visualize the same five normalized metrics—WR, CSR, improvement slope, $1-$ORR, and $1-$OBR—so that larger values uniformly indicate better behavior. DeepSeek-R1 spans the largest area, pairing a high WR with strong budget discipline (high $1-$OBR) and solid CSR, while its improvement slope is *mid-to-low*, consistent with strong openings that taper over rounds.

In Figure 5, ChatGPT-4.1 shows the *steepest improvement slope* and good budget adherence but a comparatively lower $1-$ORR, indicating steady gains alongside a greater willingness to revise. ChatGPT-4o is the most balanced mid-pack profile, with all axes around the middle to upper-middle bands. Claude-3.5-Sonnet stands out for higher CSR and decent WR yet limited slope gains. Llama-3-70B is broadly middling without a standout dimension, whereas DeepSeek-V3 traces the smallest polygon, with low WR/CSR and weaker risk and budget control. Overall, larger and more uniformly filled areas correspond to more reliable, constraint-aware reasoning, reinforcing that robust performance depends on balanced outcomes, effective revisions, and budget fidelity rather than strength on a single metric.

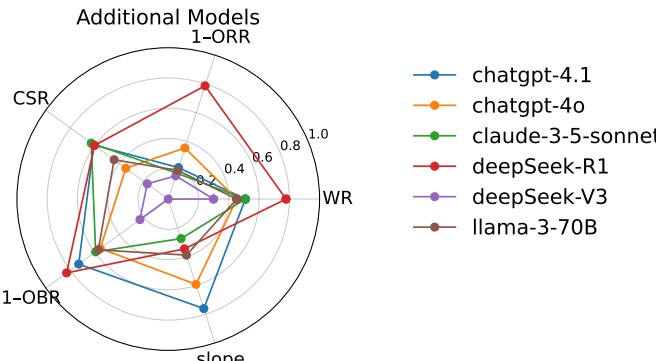

Figure 5: Radar chart of five normalized metrics for the additional-model subset: win rate (WR), correction success rate (CSR), improvement slope, $1-$over-correction risk (1–ORR), and $1-$over-budget rate (1–OBR). Higher is better on all axes.

## B  FORMAL EVALUATION METRICS

This appendix provides the rigorous mathematical formulations for all metrics used in this research. Each metric is presented using a unified three-part structure: *Definition*, *Formulation*, and *Interpretation*.

### B.1  CORE METRICS

**1. Win Rate (WR)**

**Definition.** Win Rate represents the probability that a model achieves a successful outcome while strictly adhering to the game rules.

**Formulation.** Let $\mathcal{G} = \{1, \ldots, N\}$ denote the set of all game rounds played by a model. For a given round $i$, let $r_i$ be the outcome ($r_i = 1$ for a win, $r_i = 0$ for a loss), and $v_i \in \{0, 1\}$ be the validity flag ($v_i = 1$ if all rules are satisfied).

$$\text{WR} = \frac{1}{N} \sum_{i=1}^{N} \mathbb{I}(r_i = 1 \wedge v_i = 1), \tag{1}$$

where $\mathbb{I}(\cdot)$ is the indicator function.

**Interpretation.** Any rule violation ($v_i = 0$) is treated as an automatic forfeiture ($r_i = 0$), so WR captures the fraction of rounds that are both successful *and* rule-compliant, rather than just superficially winning.

**2. Over-Correction Risk Rate (ORR)**

**Definition.** ORR is the conditional probability that a model revises its strategy after receiving negative feedback (a loss or rule violation).

**Formulation.** Let $\mathcal{F}_{\text{neg}} \subseteq \mathcal{G}$ be the subset of rounds with negative feedback. For each such round $i$, let $s_i$ denote the original strategy and $s_i'$ the strategy after feedback (if any). Define a binary function $\delta(s_i, s_i')$ that equals 1 if the model submits a revised strategy distinct from $s_i$, and 0 otherwise:

$$\text{ORR} = \frac{\sum_{i \in \mathcal{F}_{\text{neg}}} \mathbb{I}\big(\delta(s_i, s_i') = 1\big)}{|\mathcal{F}_{\text{neg}}| + \varepsilon}. \tag{2}$$

where $\varepsilon$ is a small positive constant used to avoid division-by-zero in degenerate cases.

**Interpretation.** A high ORR indicates a tendency to reactively alter strategies whenever negative feedback appears. This can reflect useful adaptivity, but in excess it may signal unstable behavior and unnecessary self-editing.

**3. Correction Success Rate (CSR)**

**Definition.** CSR measures how often a revision actually leads to an improved outcome or game state, conditional on a revision being attempted.

**Formulation.** Let $\mathcal{R} \subseteq \mathcal{F}_{\text{neg}}$ be the subset of rounds where a revision occurred (i.e., $\delta(s_i, s_i') = 1$). Let $I(s_j')$ denote the improvement condition (e.g., turning a loss or invalid state into a valid win or a strictly better state).

$$\text{CSR} = \frac{\sum_{j \in \mathcal{R}} \mathbb{I}\big(I(s_j') = \text{True}\big)}{|\mathcal{R}| + \varepsilon}. \tag{3}$$

where $\varepsilon$ is a small positive constant as above.

**Interpretation.** CSR isolates the quality of the model's feedback loop. High ORR but low CSR indicates a "spray-and-pray" correction style: frequent revisions that rarely help.

**4. Improvement Slope ($\beta$)**

**Definition.** $\beta$ captures the temporal trend of performance, measuring whether the model improves or degrades over time in a given evaluation setting.

**Formulation.** Let $y_t$ be the average win rate in time step (or batch) $t \in \{1, \ldots, T\}$, and let $\bar{t}$ and $\bar{y}$ be the sample means of $t$ and $y_t$, respectively. The ordinary least squares slope is

$$\beta = \frac{\sum_{t=1}^{T}(t - \bar{t})(y_t - \bar{y})}{\sum_{t=1}^{T}(t - \bar{t})^2}. \tag{4}$$

**Interpretation.** A positive $\beta$ indicates performance improvement over repeated interactions (effective in-context adaptation), whereas a negative $\beta$ suggests degradation or overfitting to short-horizon patterns.

**5. Over-Budget Rate (OBR)**

**Definition.** OBR quantifies how frequently a model proposes strategies that exceed explicit resource constraints (e.g., unit cost, skill cost, budget caps).

**Formulation.** Let $C(s_i)$ be the resource cost associated with strategy $s_i$, and let $B_{\max}$ denote the maximum allowed budget.

$$\text{OBR} = \frac{1}{N} \sum_{i=1}^{N} \mathbb{I}\big(C(s_i) > B_{\max}\big). \tag{5}$$

**Interpretation.** Low OBR indicates strong numerical and constraint adherence. High OBR reveals difficulty internalizing hard limits, even when they are explicitly specified in the prompt.

B.2 SUPPLEMENTARY METRICS

**6. Rule Violation Rate (RVR)**

**Definition.** RVR measures the model's failure rate with respect to explicit environment rules *at the initial planning stage*, before any feedback or revision.

**Formulation.** Let $M_i$ denote model $i$, and let $T_i$ be the total number of initial strategy proposals made by $M_i$ across all games where it provides an initial strategy. For each proposal $t$, let $S_{i,t}^{(0)}$ be the initial strategy:

$$\text{RVR}_i = \frac{1}{T_i} \sum_{t=1}^{T_i} \mathbb{I}\big(S_{i,t}^{(0)} \text{ violates rules}\big). \tag{6}$$

**Interpretation.** RVR directly reflects *constraint satisfaction capability* in the *initial planning phase*: it measures how often a model's first proposal already violates explicit rules (especially budget caps). Lower RVR means the model more reliably maps abstract constraints into valid, executable plans before any feedback is given.

### 7. Constructive Rate (CnstrR)

**Definition.** CnstrR captures how often a correction attempt leads to an objectively improved game state, distinguishing directed optimization from random perturbations.

**Formulation.** Let $M_i$ be the model under evaluation. Let $G_i$ be the total number of game instances involving $M_i$, and let $K_{i,g}$ be the number of strategies proposed by $M_i$ in game instance $g$. Let $E_{i,g,k}$ be the event that $M_i$ receives negative feedback for strategy $S_{i,g,k}$ (model $i$, game instance $g$, $k$-th strategy in that game), and let $A_{i,g,k+1}$ be the event that the model proposes a new strategy $S_{i,g,k+1}$ in response. Let $\Phi(S)$ be a game-specific state evaluation function where higher values indicate a more advantageous position. The Constructive Rate for $M_i$ is

$$
\text{CnstrR}_i = \frac{\sum_{g=1}^{G_i} \sum_{k=0}^{K_{i,g}-1} \mathbb{I}\Big(E_{i,g,k} \wedge A_{i,g,k+1} \wedge \big(\Phi(S_{i,g,k+1}) > \Phi(S_{i,g,k})\big)\Big)}{\sum_{g=1}^{G_i} \sum_{k=0}^{K_{i,g}-1} \mathbb{I}\big(E_{i,g,k} \wedge A_{i,g,k+1}\big) + \varepsilon}, \tag{7}
$$

where $\varepsilon$ is a small constant for numerical stability.

**Interpretation.** CnstrR captures whether revisions tend to make *incremental, positive progress*: a high value means that, conditional on receiving negative feedback and choosing to revise, the new state $S_{i,g,k+1}$ is more advantageous under the domain-specific heuristic $\Phi$. In other words, the model behaves more like it is performing gradient-like updates rather than issuing random perturbations that merely change the state without improving it.

### 8. First-Mover Advantage (FMA)

**Definition.** FMA quantifies how sensitive a model's performance is to moving first versus second, serving as a proxy for reasoning robustness under role asymmetries.

**Formulation.** Let $X$ be any scalar performance metric (e.g., WR, ORR, CSR). For model $M_i$, let $\mathcal{G}_{i,\text{first}}$ and $\mathcal{G}_{i,\text{second}}$ denote the sets of game instances where it moves first or second, respectively. Let $N_{i,\text{first}}(X) = |\mathcal{G}_{i,\text{first}}|$ and $N_{i,\text{second}}(X) = |\mathcal{G}_{i,\text{second}}|$ be the respective counts of such game instances for which metric $X$ is applicable. Let $X_{i,m}$ be the value of metric $X$ for model $M_i$ in game instance $m$. The average performance for $M_i$ on metric $X$ when moving first and second are

$$
\bar{X}_{i,\text{first}} = \frac{1}{N_{i,\text{first}}(X) + \varepsilon} \sum_{m \in \mathcal{G}_{i,\text{first}}} X_{i,m}, \quad \bar{X}_{i,\text{second}} = \frac{1}{N_{i,\text{second}}(X) + \varepsilon} \sum_{m \in \mathcal{G}_{i,\text{second}}} X_{i,m}, \tag{8}
$$

where $\varepsilon$ is a small positive constant to prevent division by zero if a model never plays in one of the roles or if the metric is not applicable in those instances. The First-Mover Advantage for metric $X$ and model $M_i$ is then

$$
\text{FMA}_X(i) = \bar{X}_{i,\text{first}} - \bar{X}_{i,\text{second}}. \tag{9}
$$

**Interpretation.** $\text{FMA}_X(i)$ measures the turn-order bias on metric $X$: a positive value means model $M_i$ performs better when moving first, while a negative value means it performs better as the responder. The magnitude $|\text{FMA}_X(i)|$ indicates how strongly performance depends on role; small values suggest more role-robust, generalized reasoning, whereas large values reveal heavy reliance on role-specific heuristics.

## C    USER INTERFACE SNAPSHOTS

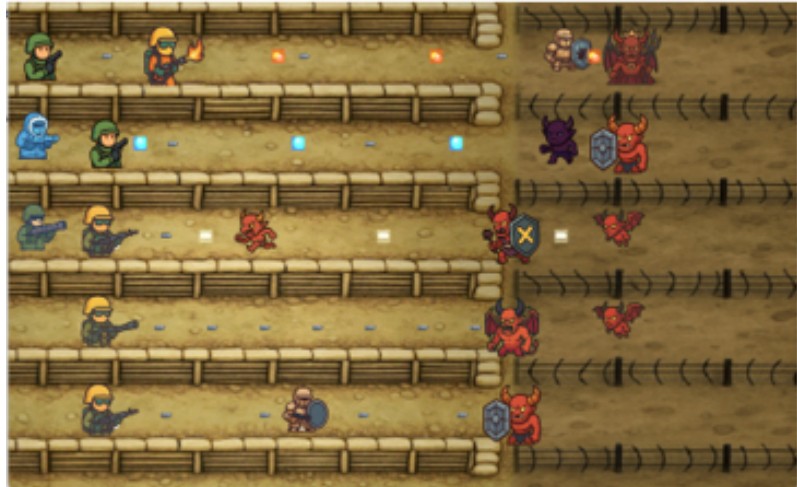

Figure 6: Tower Defense live scene. Five lanes with defender placement grids (left) and incoming demons (right). Colored orbs indicate projectile travel; lanes and columns correspond to the discrete grid used in rules and logging. This UI mirrors the A.1 rule set (rows, columns, movement, and budgeted placement).

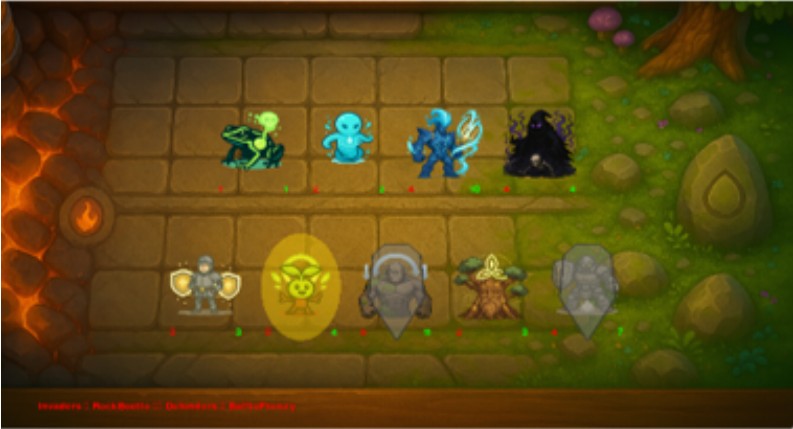

Figure 7: Battle Card preparation interface. Players select units under a resource budget before auto-resolved combat. The bench layout makes composition and role asymmetries explicit while preserving closed-loop evaluation without manual interaction.

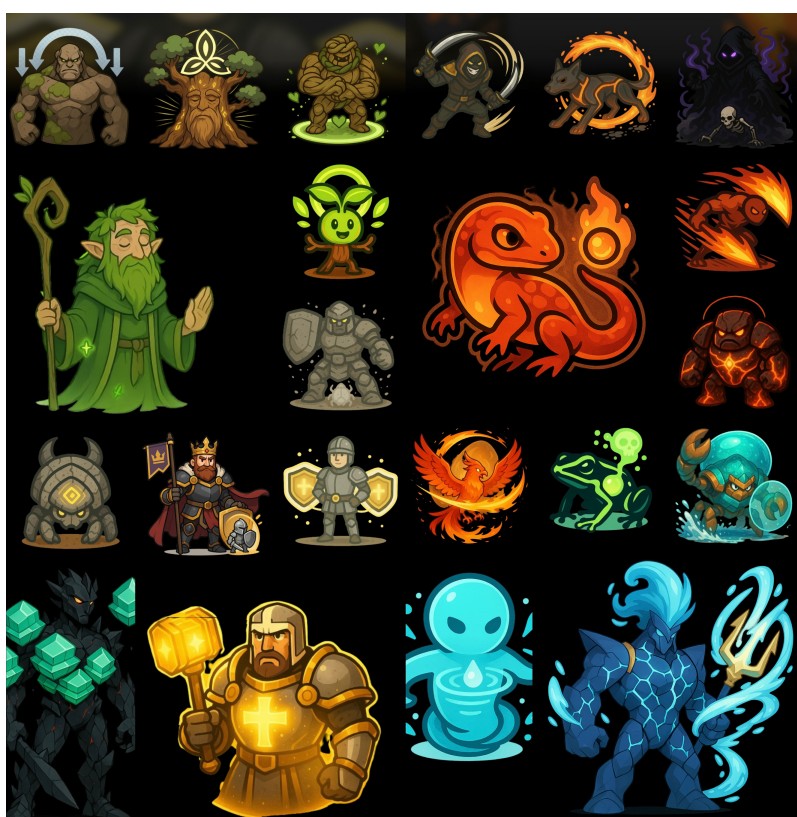

Figure 8: Attribute icons and unit art used in the Turn-based Attribute Game and Battle Card Game.

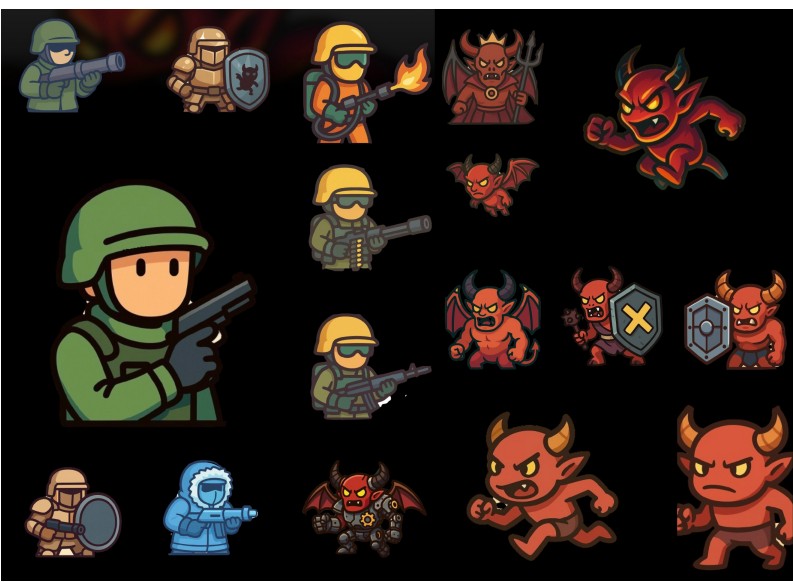

Figure 9: Tower Defense sprite sheet: human soldiers and demon variants (e.g., Shield, Machine, Bouncing, Flying).

