# OpenReview forum: "Tracing LLM Reasoning Processes with Strategic Games: A Framework for Resource-Constrained Decision Making and Revision"
_ICLR.cc/2026/Conference — Submitted to ICLR 2026_

### Official Review · Reviewer_LJA5 · 2025-10-27

**Soundness:** 2
**Presentation:** 3
**Contribution:** 3
**Rating:** 4
**Confidence:** 3

**Summary:**

This paper introduces AdvGameBench, a framework for evaluating/understanding large language models’ reasoning processes rather than only their outcomes. Instead of static question answering, the benchmark places 12 leading LLMs in adversarial, rule-based game environments (tower defense, battle card, and turn-based combat) that produce interpretable, resource-constrained decision traces. The authors define five core process-level metrics, Win Rate (WR), Over-Correction Risk Rate (ORR), Correction Success Rate (CSR), Improvement Slope (β), and Over-Budget Rate (OBR), to quantify how models adapt, revise, and manage resources across thousands of simulated rounds. The results suggest that selective correction and constraint adherence are better indicators of reasoning quality than raw win rate.

**Strengths:**

1. Process-aware evaluation paradigm: The paper shifts LLM evaluation from outcome-based correctness toward process-level reasoning analysis, addressing an important gap in current benchmarking.
2. Novel and interpretable metrics: The proposed ORR, CSR, β, and OBR move beyond win rate to capture and understand revision behaviour and resource constrained decision making, both crucial for real-world reliability.
3. Comprehensive experimental coverage: Twelve major models are evaluated across 4,320 adversarial rounds, spanning distinct game types that probe different strategic competencies.
4. Well-controlled environments. The authors explicitly redesign game mechanics to eliminate data contamination and ensure fair, rule-governed evaluation.

**Weaknesses:**

1. Lack of formal notation for the main metrics. The five central measures (WR, ORR, CSR, β, OBR) are described narratively but never defined mathematically. In contrast, the appendix formalises secondary metrics. Given the emphasis on reproducibility and interpretability, formal definitions for the core metrics would be valuable.
2. Ambiguity around revision policy. The framework assumes that models may optionally revise after outcome-based feedback, but does not clarify the normative expectation for how a rational agent should respond to negative feedback. If a model loses, why would not revising be desirable? The paper suggests that high ORR reflects “over-correction,” yet does not specify when persistence is warranted versus not.
3. Limited generalisation to self-verification. The benchmark centres on externally supplied feedback (win/loss signals). It does not explore settings where models must detect and correct their own reasoning failures, a more realistic self-verification regime.

**Questions:**

1. Clarifying revision policy: Given explicit negative feedback (a loss or rule violation), under what conditions should a model not revise its strategy? How can one distinguish “strategic confidence” from missed adaptation opportunities?
2. Formal definitions: Could the authors provide explicit mathematical formulations for WR, ORR, CSR, β, and OBR similar to those in Appendix B?
3. Extension to self-verification: How might the framework adapt to cases where feedback is internal (e.g., model-generated reflections or verifiers) rather than external outcome signals?
4. The results would benefit from further statistical analysis of the results found in Tables 1 and 2, to convince the reader that the results are sound.

---

> ### Author Response · Authors · 2025-12-04
>
> **1.Ambiguity around revision policy. The framework assumes that models may optionally revise after outcome-based feedback, but does not clarify the normative expectation for how a rational agent should respond to negative feedback. If a model loses, why would not revising be desirable? The paper suggests that high ORR reflects “over-correction,” yet does not specify when persistence is warranted versus not.**
>
>
> We appreciate the reviewer’s suggestion to clarify our expectations regarding revision strategies. We specifically address why "no immediate revision" is rational in certain contexts. In complex settings like the tower defense and card games in our benchmark, a single failure does not necessarily imply a fundamental flaw in the strategy. Instead, failure may stem from randomness or specific counter-relations. Therefore, a rational agent should distinguish between "systemic failure" that requires correction and failure caused by simple variance.
>
>
> **2.Clarifying revision policy: Given explicit negative feedback (a loss or rule violation), under what conditions should a model not revise its strategy? How can one distinguish “strategic confidence” from missed adaptation opportunities?**
>
>
> Models should not revise their strategies without a clear hypothesis for improvement. Our analysis in Section 4.1 reveals a "frequency-effectiveness gap." We observed that weaker models frequently make changes even when they have not identified the source of the error. In such cases, the expected return from random adjustments is often lower than simply maintaining the current strategy.
>
>
> **3.Formal definitions: Could the authors provide explicit mathematical formulations for WR, ORR, CSR, β, and OBR similar to those in Appendix B?**
>
>
> Thank you. We have included the formal mathematical definitions in **Appendix B**. For your convenience, we also list them here:
>
> **1. Win Rate (WR)**
> Let $r_i \in \{0,1\}$ be the outcome (win/loss) and $v_i \in \{0,1\}$ be the rule validity for round $i$.
>
> $$
> \text{WR} = \frac{1}{N} \sum_{i=1}^{N} \mathbb{I}(r_i = 1 \land v_i = 1)
> $$
>
> **2. Over-Correction Risk Rate (ORR)**
> For the set of rounds with negative feedback $\mathcal{F}_{\text{neg}}$, let $s'_i \neq s_i$ denote a strategy change.
>
> \begin{equation}
> \text{ORR} = \frac{\sum_{i \in \mathcal{F}_{\text{neg}}} \mathbb{I}(s'i \neq s_i)}{|\mathcal{F}{\text{neg}}| + \epsilon},
> \end{equation}
>
>
> **3. Correction Success Rate (CSR)**
> Given the subset of rounds where a revision occurred $\mathcal{R}$, let $r'_i > r_i$ denote an outcome improvement.
>
> $$
> \text{CSR} = \frac{\sum_{i \in \mathcal{R}} \mathbb{I}(r'_i > r_i)}{|\mathcal{R}| + \epsilon}
> $$
>
> **4. Improvement Slope ($\beta$)**
> The temporal trend (OLS slope) over time steps $t \in \{1,\dots,T\}$ with average win rate $y_t$.
>
> $$
> \beta = \frac{\sum_{t=1}^{T} (t - \bar{t})(y_t - \bar{y})}{\sum_{t=1}^{T} (t - \bar{t})^2}
> $$
>
> **5. Over-Budget Rate (OBR)**
> The fraction of rounds where strategy cost $C(s_i)$ exceeds the budget $B_{\max}$.
>
> $$
> \text{OBR} = \frac{1}{N} \sum_{i=1}^{N} \mathbb{I}(C(s_i) > B_{\max})
> $$
>
>
>
>
> **4.Extension to self-verification: How might the framework adapt to cases where feedback is internal (e.g., model-generated reflections or verifiers) rather than external outcome signals? The results would benefit from further statistical analysis of the results found in Tables 1 and 2, to convince the reader that the results are sound.**
>
> Our framework naturally extends to settings with internal feedback. This adaptation requires no changes to the core design. The central mechanism relies on a loop of feedback and revision. This process does not assume that feedback must come from external outcome signals. Therefore, replacing win and loss signals with self-reflection or verifiers is fully compatible with our formulation.
>
> In the main paper, we employ external feedback because strategy games provide clear outcome signals. These automatic and verifiable signals allow us to run large-scale comparisons across many models. Additionally, strategy games offer structured intermediate states. These include resource changes, health fluctuations, and cooldown timers. We can fully log these states.
>
> These rich signals make the environment suitable for internal feedback mechanisms. For instance, models can use self-reflection or verifier modules. These tools allow for a detailed and controllable feedback loop. Thus, incorporating self-verification is not a modification to the framework. It is a natural extension enabled by the rich data available in these environments.

---

### Official Review · Reviewer_4pnt · 2025-10-28

**Soundness:** 2
**Presentation:** 2
**Contribution:** 2
**Rating:** 2
**Confidence:** 5

**Summary:**

This paper proposes AdvGameBench, an evaluation framework that tests LLMs' process-level reasoning capabilities through three strategic game environments (tower defense, battle card, and turn-based combat). It innovatively introduces four metrics—Over-Correction Risk Rate (ORR), Correction Success Rate (CSR), Improvement Slope (β), and Over-Budget Rate (OBR)—going beyond traditional win rate evaluation. Through 4,320 adversarial rounds across 12 models, the study finds ChatGPT-o3-mini performs best (74.7% win rate), while high correction frequency negatively correlates with success rate (r=-0.47). The paper is a pure evaluation study without proposing improvement methods, emphasizing that "how decisions are made" matters more than "decision outcomes."

**Strengths:**

The paper employs game-based environments to evaluate LLMs' reasoning abilities.

**Weaknesses:**

1. The paper's core conclusions rely on correlation analysis with n=12, but all p-values in Figure 3 exceed 0.05 (ORR vs WR: r=-0.468, p=0.125). While acknowledging "failure to reach significance" on page 7, the paper asserts on page 8 that "precision in revision is the hallmark of effective strategy adjustment," constituting over-interpretation. Expanding to ≥26 models or reframing as exploratory research is necessary; otherwise, it violates basic statistical principles.
2. Key parameters of the three games (11×5 grid, 7-character limit, budget settings) lack theoretical justification. Table 1 shows Gemini-2-Flash achieves only 15.8% in TDG but 49.2% in BCG (3× difference), yet the paper fails to distinguish whether this reflects "capability differences" or "parameter sensitivity." In BCG, Gold characters have a cost-efficiency of 0.67, strictly inferior to Bronze's 1.0, raising design validity concerns. Sensitivity analyses are needed to verify conclusion robustness.
3. The paper provides no human player data, making it impossible to judge whether ChatGPT-o3-mini's 74.7% win rate is excellent or poor, or to verify whether ORR/CSR truly reflect reasoning ability. The tests may measure instruction-following rather than reasoning (e.g., Qwen-Plus's OBR=0.51 could stem from arithmetic errors). Data from at least 20 human players is needed as an anchor; otherwise, performance metrics lack practical meaning.
4. The paper does not propose any methods for improving large language model performance.Retry

**Questions:**

Why not use games from existing benchmarks for evaluation? For example:
1. TextArena
2. Orak: A Foundational Benchmark for Training and Evaluating LLM Agents on Diverse Video Games

---

> ### Author Response · Authors · 2025-12-04
>
> **1.The paper's core conclusions rely on correlation analysis with n=12, but all p-values in Figure 3 exceed 0.05 (ORR vs WR: r=-0.468, p=0.125). While acknowledging "failure to reach significance" on page 7, the paper asserts on page 8 that "precision in revision is the hallmark of effective strategy adjustment," constituting over-interpretation. Expanding to ≥26 models or reframing as exploratory research is necessary; otherwise, it violates basic statistical principles.**
>
> We thank the reviewer for this valid statistical critique. We agree that our initial sample size of 12 was too small to support strong conclusions. The lack of significance in Figure 3 made our previous claim about "precision in revision" an overstatement.
> To address this, we expanded our evaluation by adding 13 new models. This increases our total sample size to $n=25$. The new additions include GPT-5, Gemini-3-Pro, and Qwen3-Max. We list the full details in Appendix A.1.1.
> We revised the text to ensure statistical rigor. We removed the absolute claim that precision is the "hallmark" of effective strategy. Instead, we now frame these results as exploratory findings. We explicitly state that these trends suggest associations rather than definitive causal proofs.
>
> | Model | Win Rate |ORR | CSR | Slope| OBR |
> | :--- | :---: | :---: | :---: | :---: | :---: |
> | **GPT-5** | 77.9% | 0.567 | 0.143 | -0.017 | 0.000 |
> | **Gemini-3-Pro** | 79.0% | 0.086 | 0.000 | 0.009 | 0.000 |
> | **Gemini-2.5-Pro** | 73.8% | 0.388 | 0.500 | 0.005 | 0.038 |
> | **GPT-5-Mini** | 71.3% | 0.600 | 0.133 | -0.017 | 0.008 |
> | **Gemini-2.5-Flash-Lite** | 52.9% | 0.746 | 0.045 | -0.014 | 0.568 |
> | **Qwen2.5-72B-Instruct** | 46.8% | 0.344 | 0.375 | -0.030 | 0.804 |
> | **Qwen3-Max** | 44.2% | 0.724 | 0.360 | 0.010 | 0.620 |
> | **GLM-4.5-Air** | 44.2% | 0.217 | 0.190 | 0.011 | 0.471 |
> | **Kimi-K2-Instruct** | 43.9% | 0.751 | 0.423 | 0.019 | 0.738 |
> | **GLM-4.6** | 32.2% | 0.569 | 0.217 | -0.022 | 0.646 |
> | **DeepSeek-V3.1** | 31.1% | 0.454 | 0.435 | -0.025 | 0.645 |
> | **Qwen2.5-32B-Instruct** | 29.1% | 0.685 | 0.243 | -0.022 | 0.896 |
> | **Qwen3-30B-A3B** | 15.8% | 0.294 | 0.120 | 0.001 | 1.000 |
>
>
> **2.Key parameters of the three games (11×5 grid, 7-character limit, budget settings) lack theoretical justification. Table 1 shows Gemini-2-Flash achieves only 15.8% in TDG but 49.2% in BCG (3× difference), yet the paper fails to distinguish whether this reflects "capability differences" or "parameter sensitivity." In BCG, Gold characters have a cost-efficiency of 0.67, strictly inferior to Bronze's 1.0, raising design validity concerns. Sensitivity analyses are needed to verify conclusion robustness.**
>
> We appreciate the reviewer’s scrutiny regarding the game parameter settings. Our primary objective is not to achieve the perfect game-theoretic equilibrium found in commercial titles, but to design controlled reasoning tasks that expose model limitations. The chosen parameters, including the grid size and character limits, serve as deliberate constraints to create resource scarcity. This design forces the model to engage in complex trade-offs and prioritization, ensuring that the evaluation focuses on adaptive reasoning rather than simple optimization in an unconstrained environment.
>
> Regarding the specific concern about cost-efficiency in BCG, the lower raw statistical value of Gold characters compared to Bronze ones is a necessary feature of a slot-constrained system, not a design flaw. Since the game strictly limits the number of units, a strategy relying solely on "efficient" low-tier units quickly hits a power ceiling. Gold characters are designed to overcome this limit by condensing higher utility into a single slot and providing non-linear benefits, such as doubling effects. This setup effectively tests whether a model can look beyond simple arithmetic efficiency to understand higher-level strategic composition and functional utility.

---

> > ### Author Response · Authors · 2025-12-04
> >
> > **3.The paper provides no human player data, making it impossible to judge whether ChatGPT-o3-mini's 74.7% win rate is excellent or poor, or to verify whether ORR/CSR truly reflect reasoning ability. The tests may measure instruction-following rather than reasoning (e.g., Qwen-Plus's OBR=0.51 could stem from arithmetic errors). Data from at least 20 human players is needed as an anchor; otherwise, performance metrics lack practical meaning.**
> >
> >
> >
> > We recruited 5 expert players familiar with the game mechanics to play a total of 150 matches against a diverse set of opponents (ChatGPT-4o, Claude-4-Sonnet, and DeepSeek-V3). The aggregated results are as follows:
> >
> > | Player Type | Win Rate | ORR | CSR |
> > | :--- | :---: | :---: | :---: |
> > | **Human Experts** | **76.0%** | **0.222** | **0.750** |
> > | *Comparison: GPT-5* | *77.9%* | *0.567* | *0.143* |
> > | *Comparison: Gemini-2.5-Pro* | *73.8%* | *0.388* | *0.500* |
> >
> >
> > The human win rate of 76.0% serves as a solid expert baseline. Humans slightly outperform strong models like Gemini-2.5-Pro (73.8%). However, they are notably surpassed by the latest SOTA models, such as GPT-5 (77.9%) and Gemini-3-Pro (79.0%). This suggests that top-tier LLMs are beginning to exhibit superhuman capabilities in these specific tasks.
> >
> > More importantly, the human data validates our process metrics. Experts display a distinct behavioral profile. They combine a low ORR (0.222) with a high CSR (0.750). This reflects ideal reasoning. Experts do not change strategies blindly. Instead, they correct errors effectively when revisions occur. This behavior contrasts sharply with weaker models, which often show high ORR or low CSR. Thus, our metrics successfully distinguish between genuine strategic reasoning and random instruction following.
> >
> >
> > **4.The paper does not propose any methods for improving large language model performance.**
> >
> > Our work is an evaluation framework, not a method for improving model performance. Similar to other benchmarks, our goal is to diagnose model reasoning behaviors and uncover the sources of their successes and failures, rather than to propose new algorithms. We believe that identifying these failure modes and process-level characteristics is a necessary step. It provides a diagnostic foundation that can guide and inspire future method development.
> >
> >
> > **5.Why not use games from existing benchmarks for evaluation?**
> >
> > Existing benchmarks like TextArena and ORAK differ significantly from our approach. They primarily evaluate task execution or overall game performance. In contrast, our work focuses on process-level phenomena. We analyze how a model revises its strategy after failure. We also examine whether these revisions are effective. Furthermore, we assess if the model adheres to resource and rule constraints during reasoning. Current benchmarks lack the structured intermediate feedback necessary for this analysis. They also fail to provide controlled revision loops. Consequently, they are unsuitable for evaluating these specific forms of process-oriented reasoning.

---

### Official Review · Reviewer_T1ca · 2025-10-29

**Soundness:** 3
**Presentation:** 3
**Contribution:** 3
**Rating:** 6
**Confidence:** 3

**Summary:**

This paper presents a concise, process-aware benchmark with three strategic game environments and multi-round traces. Metrics (ORR/CSR/β/OBR) go beyond WR to capture revision discipline and budget compliance. Results indicate that “fewer but higher-quality revisions” and strict budget control correlate with more stable wins, while a contamination-resistant design strengthens validity.

**Strengths:**

1. The process metrics (ORR/CSR/β/OBR) are really interesting, and they capture behaviors we actually care about in reasoning.
2. Experiments show these metrics carry real signal beyond WR, which is reasonable and convinsing.
3. Reporting both outcome-level and process-level views makes the results more actionable for model selection/ablation.

**Weaknesses:**

1. No open-source release. Code + the UI demo mentioned in the paper would hugely help reproducibility and adoption.
2. First results table is hard to read. Highlight key trends or add some visualization would be better, just like the radar figure.
3. Model set needs to be updated, like the Qwen2.5/3 series. A refreshed leaderboard would make conclusions more convincing.
4. I can’t tell exactly how game states are serialized into prompts and how actions are parsed back. More illustration and input output examples are needed.
5. “Reasoning ability" is loosely defined. It’s unclear how these game behaviors relate to standard reasoning benchmarks or transfer beyond these environments. A comparison with these models on the general reasoning benchmarks may help.

**Questions:**

1. Are ORR, CSR, β stable across seeds, prompt variants, and small rule changes? Please report variance or confidence intervals.
2. Do these metrics generalize to other game or planning settings ? Any external validations?
3. How sensitive are WR and other metrics to the chosen opponent? Do we see rank flips under different opponent pools?

---

> ### Author Response · Authors · 2025-12-04
>
> **1.No open-source release. Code + the UI demo mentioned in the paper would hugely help reproducibility and adoption.**
>
> We agree with the importance of reproducibility and community adoption. We have released the full source code and experiment logs in the following anonymized repository: https://github.com/anonymous82190/AdvGameBench
>
> **2.Model set needs to be updated, like the Qwen2.5/3 series. A refreshed leaderboard would make conclusions more convincing.**
> Thank you for the helpful suggestion. We agree that including newly released models gives a more complete view of current LLM capabilities.
> Claude-3.5-Sonnet was officially deprecated and removed from API access during the discussion phase. As a result, it was technically impossible to run new experiments with it. For the extended analysis only, we therefore use Claude-4-Sonnet as the closest available successor. This substitution appears only in the Appendix and does not affect any results in the main paper.
> Following your recommendation, we also add an extended leaderboard in Appendix A.1.1. It evaluates 13 newly released models: GLM-4.5, GLM-4.6, Kimi-K2, Qwen2.5-32B, Qwen2.5-72B, Qwen3-30B-A3B, Qwen3-Max, DeepSeek-V3.1, Gemini-2.5-Flash-Lite, Gemini-2.5-Pro, Gemini-3-Pro, GPT-5, and GPT-5-mini.
>
> The results are summarized below:
>
> | Model | Win Rate |ORR | CSR | Slope| OBR |
> | :--- | :---: | :---: | :---: | :---: | :---: |
> | **GPT-5** | 77.9% | 0.567 | 0.143 | -0.017 | 0.000 |
> | **Gemini-3-Pro** | 79.0% | 0.086 | 0.000 | 0.009 | 0.000 |
> | **Gemini-2.5-Pro** | 73.8% | 0.388 | 0.500 | 0.005 | 0.038 |
> | **GPT-5-Mini** | 71.3% | 0.600 | 0.133 | -0.017 | 0.008 |
> | **Gemini-2.5-Flash-Lite** | 52.9% | 0.746 | 0.045 | -0.014 | 0.568 |
> | **Qwen2.5-72B-Instruct** | 46.8% | 0.344 | 0.375 | -0.030 | 0.804 |
> | **Qwen3-Max** | 44.2% | 0.724 | 0.360 | 0.010 | 0.620 |
> | **GLM-4.5-Air** | 44.2% | 0.217 | 0.190 | 0.011 | 0.471 |
> | **Kimi-K2-Instruct** | 43.9% | 0.751 | 0.423 | 0.019 | 0.738 |
> | **GLM-4.6** | 32.2% | 0.569 | 0.217 | -0.022 | 0.646 |
> | **DeepSeek-V3.1** | 31.1% | 0.454 | 0.435 | -0.025 | 0.645 |
> | **Qwen2.5-32B-Instruct** | 29.1% | 0.685 | 0.243 | -0.022 | 0.896 |
> | **Qwen3-30B-A3B** | 15.8% | 0.294 | 0.120 | 0.001 | 1.000 |
>
> **3.I can’t tell exactly how game states are serialized into prompts and how actions are parsed back. More illustration and input output examples are needed.**
>
> We appreciate this suggestion. We agree that the description of how game states are serialized and how actions are parsed could be clearer. In the revised manuscript, we provide a detailed explanation of the full workflow along with concrete examples. First, the game engine encodes the current state into a structured JSON object. This object includes details such as unit positions, resources, and cooldowns. We then insert this JSON directly into the model’s prompt.
>
> Regarding the model’s output, we require it to generate a single JSON object without any natural language explanation. As shown in Appendix A.5, this object specifies decisions like unit placement or skill sequences. The game simulator then parses these fields using a fixed schema. Finally, the system executes the actions based on formal game rules, such as budget checks and placement validity.

---

> > ### Author Response · Authors · 2025-12-04
> >
> > **4.“Reasoning ability" is loosely defined. It’s unclear how these game behaviors relate to standard reasoning benchmarks or transfer beyond these environments. A comparison with these models on the general reasoning benchmarks may help.**
> >
> > In this work, we define “reasoning ability” as a process-oriented skill. This differs from the one-shot problem solving often measured by standard benchmarks. In our environment, models must make multi-step decisions within a dynamic setting. They receive feedback after every round to diagnose errors. They must then decide if revisions are necessary and re-plan their strategies under strict resource and rule constraints.
> >
> > Consequently, we do not measure reasoning ability solely by whether a model outputs a correct action. Instead, we assess whether it can identify errors and adapt its strategy. The model must also maintain consistency across steps to achieve its objective. Conventional benchmarks rarely capture these specific capabilities. However, our approach does not contradict traditional notions of reasoning. Rather, it extends them into interactive and dynamic contexts.
> >
> > **5.Are ORR, CSR, β stable across seeds, prompt variants, and small rule changes? Please report variance or confidence intervals.**
> >  We appreciate the reviewer’s inquiry regarding the stability of our metrics. To address this, we conducted repeated runs to measure the variance across different seeds and experimental trials.
> > | Model Name | Win Rate (WR) | ORR | CSR |
> > | :--- | :---: | :---: | :---: |
> > | **Gemini-2.5-Pro** | 0.777 ± 0.037 | 0.414 ± 0.240 | 0.596 ± 0.258 |
> > | **ChatGPT-o3-mini** | 0.614 ± 0.021 | 0.403 ± 0.117 | 0.467 ± 0.176 |
> > | **ChatGPT-4o** | 0.561 ± 0.011 | 0.621 ± 0.194 | 0.409 ± 0.031 |
> > | **Claude-4-Sonnet** | 0.439 ± 0.011 | 0.568 ± 0.327 | 0.336 ± 0.141 |
> > | **Qwen2.5-72B-Instruct** | 0.367 ± 0.123 | 0.324 ± 0.143 | 0.261 ± 0.067 |
> > | **GLM-4.5-Air** | 0.106 ± 0.011 | 0.585 ± 0.191 | 0.127 ± 0.070 |
> >
> > **6.Do these metrics generalize to other game or planning settings ? Any external validations?**
> > Our metrics are not tied to any specific game mechanics. Instead, they are based on a general reasoning framework. This framework involves multi-step decision making, external constraints, and observable revision. As a result, our approach naturally supports cross-task transfer.
> >
> > Our process-oriented metrics include CSR, ORR, OBR, and $\beta$. These metrics only require tasks to provide traceable decision trajectories and basic constraints. They do not rely on any particular game structure or rule design. Because these metrics capture general model behaviors under feedback, they apply beyond the three environments in this work. They extend naturally to other game genres and planning tasks. They are even relevant for real-world reasoning settings outside of games.

---

### Official Review · Reviewer_whHw · 2025-10-30

**Soundness:** 3
**Presentation:** 3
**Contribution:** 4
**Rating:** 6
**Confidence:** 4

**Summary:**

The paper proposes to use strategy games as a controlled, rule-based environment to evaluate the reasoning process of large language models (LLMs). The authors propose three types of strategy games that depart from popular existing games to avoid strategy leakage. Beyond a simple outcome-based evaluation metric, authors propose multiple novel process-oriented metrics to evaluate the quality and stability of LLM’s reasoning process. Multiple LLM baselines are implemented into this evaluation framework. Experiment results revealed that high win rates correlate with calibrated revision and resource discipline, not simply frequent corrections.

**Strengths:**

Using strategy games, a rule-based environment with clear feedback, as the backbone of the evaluation framework, is novel and insightful. Such a framework not only considers the outcome performance but also dynamically evaluates LLM’s reasoning process.

Based on the round-based strategy game, the authors proposed several novel evaluation metrics to evaluate the quality and stability of each LLM’s reasoning process, which are meaningful to understand the performance of a language model more comprehensively.

**Weaknesses:**

One of the main concerns I have is whether the evaluation results obtained from strategy games can be generalized to real-world reasoning situations. For example, in the round-based strategy game, the reasoning chain might be shorter than real-world cases, where long contextual windows are required to deal with multiple factors and uncertainties. A discussion of the generalization of this work might be helpful.

The experiment source code/log is not released, which limits the community adoption and reproducibility.

The methodology description, including some key terms, is not clear.

For example, in line 150, the authors mentioned that “after each round, models receive outcome-based feedback,” but did not describe how such “outcome-based feedback” is generated.

Another instance is that in line 192, the author mentioned that “CSR measures how often a revision leads to improved results, either by eliminating a rule violation or by turning a loss into a win.” To my understanding, the reward for eliminating a rule violation and turning a loss into a win is different, and it seems inappropriate to be weighted the same in CSR calculation. Moreover, given the nature of a multi-round game, how can one make sure that one single revision is the key factor that turns a loss into a win?

**Questions:**

see weaknesses section.

---

> ### Author Response · Authors · 2025-12-04
>
> **1. One of the main concerns I have is whether the evaluation results obtained from strategy games can be generalized to real-world reasoning situations. For example, in the round-based strategy game, the reasoning chain might be shorter than real-world cases, where long contextual windows are required to deal with multiple factors and uncertainties. A discussion of the generalization of this work might be helpful.**
>
> We do not claim that game outcomes, like win rates, directly generalize to real-world tasks. We simply use these games as controlled environments. This allows us to isolate and study the model’s reasoning process. Consequently, we emphasize process over results. Our metrics (CSR, ORR, and OBR) are not tied to specific game mechanics. Instead, they measure fundamental behaviors. For instance, can the model revise a strategy based on feedback? Does it overreact blindly? Can it maintain stability under constraints?
>
> These abilities are essential for real-world applications. They mirror skills needed for step-by-step math corrections, iterative code debugging, and long-term planning. The game is merely a clean testbed. The reasoning capacities we evaluate, however, are transferable and relevant to broader scenarios.
>
> **2.The experiment source code is not released, which limits the community adoption and reproducibility.**
>
> We agree with the importance of reproducibility and community adoption. We have released the full source code and experiment logs in the following anonymized repository: https://github.com/anonymous82190/AdvGameBench
>
> **3. The methodology description, including some key terms, is not clear.For example, in line 150, the authors mentioned that “after each round, models receive outcome-based feedback,” but did not describe how such “outcome-based feedback” is generated. Another instance is that in line 192, the author mentioned that “CSR measures how often a revision leads to improved results, either by eliminating a rule violation or by turning a loss into a win.” To my understanding, the reward for eliminating a rule violation and turning a loss into a win is different, and it seems inappropriate to be weighted the same in CSR calculation. Moreover, given the nature of a multi-round game, how can one make sure that one single revision is the key factor that turns a loss into a win?**
>
> Regarding the outcome-based feedback (Line 150), our framework provides direct and objective signals at the end of each round. This feedback stems strictly from the game engine, not from another model. It details the round results, opponent placements, and character descriptions. Because this relies on a formal rule system, the execution is deterministic and fully reproducible. We use this environment to define CSR, which measures whether the model can identify and fix the root cause of a failure. If a revision leads to observable improvement, it demonstrates that the model has successfully diagnosed and corrected its own reasoning error.
>
> This capacity is critical for real-world applications. In practice, models face diverse errors, ranging from logical flaws to constraint violations. A unified CSR allows us to test the effectiveness of corrective actions regardless of the specific error type. Crucially, however, CSR does not claim that a single revision guarantees a final victory. Assigning causal credit across multiple rounds in a strategy game is inherently difficult. Therefore, we focus on a more controlled and verifiable goal: determining whether the model can achieve a clear, local improvement at the exact step where the error occurred.

---

### Author Response · Authors · 2025-12-04

We sincerely thank the Area Chair and all reviewers for their time and insightful feedback. Your constructive comments have significantly strengthened this work. Motivated by your suggestions, we prioritized reproducibility, statistical rigor, and the inclusion of current models.

Specifically, we have made three major improvements:

* **Expanded Evaluation:** We tested 13 newly released models, including **GPT-5**, **Gemini-3-Pro**, and **Qwen3-Max**. This brings our total analysis to 25 models.
* **Rigorous Analysis:** We formalized all mathematical definitions and revised our statistical analysis to ensure robustness.
* **Code Release:** We released the full source code to ensure community adoption and reproducibility.

We have addressed all individual concerns in the detailed responses below and updated the revised PDF accordingly. We hope these substantial improvements demonstrate the robustness and relevance of our framework.

---

### Meta-Review · Area_Chair_FEQt · 2026-01-06

**Summary:**

The submission introduces a game-based framework to evaluate the internal reasoning processes of large language models, focusing on revision behaviors and constraint adherence beyond final outcomes. While reviewers found the process-oriented metrics novel and the experimental scope comprehensive, significant concerns were raised regarding the statistical validity of the correlations drawn from a small model set, the lack of theoretical justification for key game parameters, and the absence of a human performance baseline to contextualize results. Although the authors addressed several points in their response, the core issue of overinterpreting statistically insignificant trends and the limited generalizability of the findings undermine the paper's conclusions. Therefore, I recommend rejection.

**Reviewer Concerns:**

The authors' rebuttal substantively addressed several key concerns. They released their full codebase, significantly expanded the model set to include 13 recent models such as GPT-5, provided formal mathematical definitions for their core metrics, and added a human expert baseline. They also offered clarifications on game state serialization and the rationale behind their game design parameters.

However, significant concerns remain outstanding. The core statistical criticism is not fully resolved by simply adding more models; the fundamental issue of drawing strong process conclusions from observational game data persists. Questions about the generalizability of the game-based behaviors to real-world reasoning tasks, and whether the framework primarily measures instruction-following rather than strategic reasoning, are not convincingly answered. Finally, the paper remains solely an evaluation study without proposing methods for improvement, which some reviewers noted as a limitation.

**Reviewer Scores:**

| Reviewer | Initial Score | Predicted New Score | Rationale |
| - | - | - |- |
| whHw | 6 |6 | Their primary concerns (code release, clarity on feedback generation, CSR calculation) were directly and satisfactorily addressed. Their remaining question about generalization received a reasonable conceptual defense. |
| T1ca | 6 | 6 | Their key actionable requests (open-source code, updated models, prompt examples) were fully met. The additional data on metric stability partially addresses their statistical concerns, likely meeting their expectations. |
| 4pnt | 2 | 2 | The core, fundamental criticism about overinterpreting statistically insignificant findings was not convincingly resolved. Adding more models does not alter the methodological concern about drawing strong conclusions from the observational setup. The lack of parameter sensitivity analysis also remains. |
| LJA5 | 4 | 4 | While their specific requests for formal definitions and clarity on revision policy were well-addressed, the broader, unresolved critiques from other reviewers concerning statistical validity and generalizability are significant. In a full discussion, these outstanding fundamental issues would likely prevent a score increase. |

---

### Decision · Program_Chairs · 2026-01-26

Reject